# Nanoparticle Effects on Stress Response Pathways and Nanoparticle–Protein Interactions

**DOI:** 10.3390/ijms23147962

**Published:** 2022-07-19

**Authors:** Shana J. Cameron, Jessica Sheng, Farah Hosseinian, William G. Willmore

**Affiliations:** 1Department of Chemistry, Carleton University, Ottawa, ON K1S 5B6, Canada; shana.cameron@carleton.ca (S.J.C.); farah.hosseinian@carleton.ca (F.H.); 2Department of Biology, Carleton University, Ottawa, ON K1S 5B6, Canada; jessica.sheng@carleton.ca; 3Institute of Biochemistry, Carleton University, Ottawa, ON K1S 5B6, Canada

**Keywords:** nanoparticles, oxidative stress, immune system, inflammation, mitochondrial function, detoxification enzymes, insulin signaling, calcium signaling, nitric oxide signaling, nanoparticle–protein interactions

## Abstract

Nanoparticles (NPs) are increasingly used in a wide variety of applications and products; however, NPs may affect stress response pathways and interact with proteins in biological systems. This review article will provide an overview of the beneficial and detrimental effects of NPs on stress response pathways with a focus on NP–protein interactions. Depending upon the particular NP, experimental model system, and dose and exposure conditions, the introduction of NPs may have either positive or negative effects. Cellular processes such as the development of oxidative stress, the initiation of the inflammatory response, mitochondrial function, detoxification, and alterations to signaling pathways are all affected by the introduction of NPs. In terms of tissue-specific effects, the local microenvironment can have a profound effect on whether an NP is beneficial or harmful to cells. Interactions of NPs with metal-binding proteins (zinc, copper, iron and calcium) affect both their structure and function. This review will provide insights into the current knowledge of protein-based nanotoxicology and closely examines the targets of specific NPs.

## 1. Introduction

Nanoparticles (NPs) are small particles with at least one dimension that is 1–100 nm in length and can be any shape. Nanoparticles are currently used in a wide and ever-increasing number of commercial, agricultural, medical, and research applications due to their usefulness in these areas; however, the biological effects of exposure to these NPs are not well understood. Nanoparticle exposure may also occur due to their presence as by-products in substances such as engine exhaust [1,2], cigarette smoke [3], electronic cigarette fluid and aerosols [4,5], cooked food [6], and various spray products [7]. The most detrimental exposure route for significant levels of NPs is through inhalation, a situation that can especially occur during workplace manufacturing, or through exposure to exhaust fumes or sprays [8,9]. Additional exposure routes are through skin contact (e.g., creams, sunscreens, and cosmetics) and ingestion (e.g., food colorants and food packaging). Less commonly, NPs may be administered intravenously such as for medicinal purposes or as part of drug delivery systems [10,11].

It is worth noting that the physical properties and biological effects of the nanoparticle form of a material may be different than that of the bulk material. This is due to several factors including the extremely small size of the NPs, increased surface area to volume ratio, increased potential for surface reactivity (depending on the material), ease of access into cells, and increased interaction with cellular components. In addition, since the smaller NPs have fewer atoms, the energy levels may become more discrete, and the energy band gap may become wider. This potentially changes the electrical conductivity as well as the absorption spectra of the nanomaterial [8]. In biological systems, electron transfer may occur between NPs and the cellular components if the conduction band of the NP coincides with the redox potential of the cellular reactions (approximately −4.12 to −4.84 eV). This situation could result in the occurrence of cellular oxidation-reduction reactions and the production of oxidative stress [8]. This may explain why many NPs produce oxidative stress in biological systems. For example, titanium dioxide (TiO_2_) NP toxicity is thought to be mainly triggered by oxidative stress [12]. Additionally, for metallic nanoparticles such as silver (Ag) NPs or iron oxide NPs, oxidative stress may be caused by the ions that are released from the NP surface, with smaller NPs releasing more ions due to their higher surface area to volume ratio [12]. Thus, especially for smaller NPs, oxidative stress in biological systems due to NP exposure is thought to be one of the main factors in nanotoxicity [13,14]. Oxidative stress activates stress response pathways and other signaling cascades. Higher levels of oxidative stress may result in inflammation, cellular damage, lipid peroxidation, deoxyribonucleic acid (DNA) damage, and apoptosis. In addition to size, factors such as shape, surface coating, charge, solubility, and aggregation state all contribute to or mitigate the overall effects and toxicity of the NP [14].

Proteins are known to adsorb onto the NP surfaces in biological environments due to the high surface free energy. This decreases the surface free energy and stabilizes the NP. The adsorbed proteins can bind strongly and persistently onto the NP surface to form a hard corona, or they may form a soft corona and bind transiently to either the NP surface or to the hard corona. These protein coronas may alter the biological distribution, effects, and toxicity of the NP [15,16]. Especially for larger NPs and micron-sized particles that can adsorb a substantial number of biomolecules due to their large surface area, the cellular effects of the particles themselves may be less than the cellular effects of their surface biomolecules [8].

### 1.1. Interactions of Nanoparticles with Cellular Components

Smaller NPs (<100 nm) are taken into cells via endocytosis or through diffusion across the cell membrane. Larger particles (>100 nm) are generally too large to be taken into endosomes and may instead be taken up by phagocytosis [8]. Once inside the cell, specific cellular interactions may occur between the NPs and the cellular components such as molecular substitution, protein conformational change, and protein dysfunction [15,17]. Smaller NPs have greater access and ability to interact with the cellular components than larger NPs [8]. Ultrasmall NPs (which are defined as being 1–10 nm) are biocompatible and can mimic biological molecules in the cell, bind to cellular receptors, and initiate signaling [8]. For example, ultrasmall silica (Si) NPs (3.6 ± 0.5 nm) have been found to directly bind to the T cell receptor complex and activate T cells in the immune system and their downstream signaling. This direct interaction between the ultrasmall Si NPs and the T cell receptor complex was determined using competitive binding experiments, with molecular modeling to support that the interaction was theoretically possible both sterically and electrostatically [18,19]. Interactions between proteins and NPs may also cause protein conformational changes, loss of structure, and dysfunction [15,17,20]. An example of this has been demonstrated with cytochrome c (Cyt c) and TiO_2_ nanocables. These nanocables were formed from nanowires 3–4 nm in width that self-assembled into flat cables with a collective width of 70–130 nm and length of 400–800 nm. Experiments using surface enhanced infrared absorption spectroscopy (SEIRAS) and electrochemical cyclic voltammetry indicated that interaction between the TiO_2_ nanocables and Cyt c resulted in protein unfolding and reduced binding with its binding partner mimic, 11-mercaptoundecanoic acid [15]. Additionally, 30 nm silicon dioxide (SiO_2_) NPs have been found to increase the exposure of hydrophobic groups in α-synuclein, leading to increased α-synuclein aggregation and the formation of β-amyloid fibrils, with this being demonstrated using circular dichroism (CD) spectroscopy, transmission electron microscopy (TEM) imaging, 8-anilino-1-naphthalene sulfonate fluorescence, and Congo red absorbance [21]. The β-amyloid fibrils prepared by co-incubation of α-synuclein and SiO_2_ NP resulted in increased cytotoxicity, and triggered mitochondrial mediated apoptosis in human neuroblastoma SH-SY5Y cells compared to treatment with β-amyloid fibrils prepared without SiO_2_ NPs [21]. Furthermore, molecular docking studies using CHIMERA and PyMOL revealed that the SiO_2_ NPs bind to the N-terminus of α-synuclein through hydrogen and hydrophobic bonds, with the N-terminus being the site for membrane binding and membrane induced helix formation in the protein [22].

Even if NPs are not brought into the cell (such as may be the case for larger NPs or aggregates of NPs), they may still initiate receptor-mediated signaling cascades and cause oxidative stress. An example of this is seen with 100 nm Ag NPs, which were found to mainly stay on the surface of LoVo human colon carcinoma cells and activated p21-activated kinase (PAK), mitogen-activated protein kinase (MAPK), and phosphatase 2A signaling pathways [23].

This review aims to examine some of the specific effects of NPs on the main cellular systems and on stress response pathways, as well as their interactions with specific proteins (Figure 1).

### 1.2. Nanoparticles and Their Applications

Nanoparticles can be composed of various materials such as inorganic elements, polymers, lipids, hydrogels, carbon nanoparticles, and quantum dots [10,13,16]. Figure 2 summarizes the classes of nanoparticles and their common applications. Depending on the desired function, combinations of different materials are used for NP surface coatings and the base NP, with these also affecting the overall NP toxicity [24].

There are many different NPs that are used in the studies in this review. This is due to the fact that there is a wide variety of applications for NPs (see Table 1). Metal-based NPs are the most commonly used NPs with a broad variety of applications. Platinum (Pt) NPs have useful thermoplasmonic properties and are a good catalyst. As such, they have been added as a catalyst in fuel, used as an electrocatalyst, and are also used in cosmetics, electronic devices, and various sensors. Platinum NPs are also being used in medical implants, drug delivery systems, and photothermal therapy [25,26]. Silver NPs are the most commonly found NP in consumer products [27]. They have many commercial, medical, and agricultural applications due to their excellent antibacterial, antiviral, antifungal, antiparasitic, anticancer, and photosensitive properties, and are found in food packaging, antimicrobial clothing, cosmetics, electronics, medical devices, and bandages [24,28,29]. Gold NPs are stable, bind easily to amine or thiol compounds for surface modifications, are generally non-toxic, and are used in a wide variety of sensors, drug delivery systems, bioimaging systems, and photothermal therapy [13,24,30,31,32].

Metal oxide-based NPs of different compositions are used for a broad variety of applications. Titanium dioxide NPs are widely used for their brilliantly white pigmentation and transparent properties in paints, paper, plastics, cosmetics, toothpaste, transparent films, food packaging, construction materials, and for UV light blockage in sunscreen. Used as a food additive in items such as chewing gum and candy, TiO_2_ is labeled as food coloring agent E171 and is comprised of 25–40% nano-sized particles with the rest being micro-sized [17,33,39,42,43,44]. In medicine, TiO_2_ NPs are used in imaging and drug delivery systems [13]. Additionally, biomedical implants made using titanium alloys have been found to release TiO_2_ NPs into the surrounding tissue [45]. Zinc oxide NPs are commonly used in sunscreens, cosmetics, pigments, paints, UV light detectors, gas sensors, and in agriculture as a nanofertilizer [24,34,48]. Silicon dioxide NPs have useful optical properties, are biocompatible, and have a high surface adsorption capacity [16]. As such, they are widely used in cosmetics, as a food additive, for drug delivery systems, in biosensors, in agricultural applications, to reinforce rubber, and to improve the characteristics of concrete and mortar [17,24,40,41]. Iron oxide NPs are commonly used in drug delivery systems, biomedical applications, and bioimaging systems such as magnetic resonance imaging [13,24]. Aluminum oxide (Al_2_O_3_) NPs are widely used in paints, textiles, construction materials, polymers, biomaterials, and in fuel cells [13,24,33]. Copper oxide (CuO) NPs are used for their electrical properties in semiconductors and heat transfer fluids, as well as in coatings on medical devices due to their antimicrobial properties [24]. Cerium dioxide (CeO_2_) NPs undergo redox cycling and transition between the Ce^+3^ and Ce^+4^ oxidation states. They are used as a fuel additive for their catalytic activity and are also used in various biomedical applications due to their anti-inflammatory, antioxidant, and antimicrobial properties [35,36].

Quantum dots are 2–10 nm semiconductor crystals that are inherently fluorescent. Their absorption onset and emission fluorescence can range from violet to deep red depending upon their diameter. The larger the quantum dot, the redder (lower energy) its absorption onset and fluorescence spectrum, while smaller quantum dots absorb and emit bluer (higher energy) light. The cores of quantum dots are commonly comprised of elements such as cadmium, selenium, tellurium, zinc, indium, and silicon [46,47]. Quantum dots are used in nanomedicine as fluorescent labels in imaging and diagnostic tests, and for drug delivery systems [16].

Carbon NPs are the most used nanomaterial in drug delivery systems [13]. Carbon black NPs are mostly composed of elemental carbon, have a high surface reactivity, and have many applications such as pigments, cosmetics, or as an important reinforcing agent in rubber [34,49,50]. Fullerenes are an allotrope of carbon and are used for their remarkable ability to bind to biomolecules and distribute into the cells in biological organisms. Tube shaped fullerenes are termed carbon nanotubes and have excellent electrical conductivity as well as being used in construction materials [13,33]. Graphene forms flat two-dimensional (2D) sheets of carbon atoms that are arranged in a hexagonal pattern. Graphene oxide NPs are oxidized graphene that have hydroxyl, epoxy, and carbonyl groups attached and are useful in various applications such as drug delivery systems, imaging systems, and photothermal therapy [37].

Polymeric NPs and dendrimers are very adaptable to different applications. Since they do not generally cause a strong immune response, they are biocompatible and useful for applications in nanomedicine such as drug delivery systems, diagnostic tests, and in vaccine production [11,13,51]. Their biodegradability results in a controlled release of the drug, protein, or DNA that they are carrying, as well as a decrease in the toxicity [10,16]. Commonly used polymers include natural polymers (such as chitosan, albumin, gelatin, starch, cellulose, and hyaluronic acid), synthetic polymers (such as polyethylene glycol (PEG), polyvinyl alcohol (PVA), and polylactic acid (PLA)), and synthetic copolymers (such as poly(lactide-co-glycolide) (PLGA)) [10,16,51]. Liposomes are used in cosmetics as well as in nanomedicine for drug delivery systems and vaccine production [11,13,34].

Nanoplastics and microplastics are produced for industrial purposes or are created through the degradation of plastic into tiny particles [38]. The size of microplastic particles has not been officially defined; however, this term is generally used for particles that are 0.1–5000 µm, while nanoplastic particles are generally 1–100 nm. The most commonly used plastics include polypropylene, polyethylene, polyvinyl chloride, polyurethane, polyethylene terephthalate, and polystyrene. Nanoplastics and microplastics are emerging and concerning pollutants and are becoming increasingly present in the air, water, and soil due to the high use and improper disposal of plastics. Additionally, people are being increasingly exposed to microplastics and nanoplastics through plastic products and contamination in food [38,52].

## 2. Effects of Nanoparticles on Oxidative Stress and Stress Response Pathways

It is now currently established that engineered nanoparticles cause oxidative stress by generating harmful reactive oxygen species (ROS) [53,54]. The physicochemical properties of nanoparticles including size, shape, structure, and elemental constituents (including the presence of metals) contributes to their ability to generate ROS [55]. Toxic levels of ROS lead to the development of pathophysiological effects including DNA damage and genotoxicity, inflammation, and fibrosis [56,57,58,59]. These processes, in turn, can lead to the development of cancer [60,61], atherosclerosis [62], neurodegenerative diseases [63,64], autoimmune diseases [65,66], and diabetes mellitus [67,68]. While the implications of nanoparticle-induced oxidative stress and the development of disease have been the focus of other reviews, this review will examine the direct interactions of nanoparticles with components of various stress pathways. Table 2 summarizes the effects of NPs on oxidative stress and stress response pathways.

Many antioxidant enzymes are dependent upon metals for their catalytic activities. This includes catalase (CAT), the superoxide dismutases (SODs), and the glutathione peroxidases (GPxs). Others, such as glutathione reductase (GR), the peroxiredoxins (PRDXs), thioredoxin (TRX), and glutaredoxin have redox-sensitive cysteines at their catalytic sites that can be disrupted by metal-binding. The catalytic mechanisms of both types of antioxidant enzymes can be disrupted by metal containing nanoparticles. The catalytic activity of metal-dependent antioxidant enzymes may be disrupted by the substitution of nanoparticle-derived metals at their active sites. Catalase is an antioxidant enzyme that catalyzes the conversion of hydrogen peroxide (H_2_O_2_) to oxygen (O_2_) and water (H_2_O) and is found in nearly every organism from bacteria to humans. Catalase is one of the fastest enzymes known, with the highest turnover numbers of all known enzymes (40,000,000 molecules/s) [81]. The enzyme is a tetramer with an iron-bound heme covalently linked to each polypeptide chain. Metals, derived from metal-based nanoparticles, can inhibit not only the function of heme dependent enzymes, such as CAT and cytochrome P450s [80], but also the synthesis of heme itself [82]. In one study in lymphocytes, zero valent iron (ZVFe) NP interaction with human hemoglobin (Hb) resulted in heme displacement and degradation and induction of protein carbonylation, a measure of protein damage due to ROS [80]. In fish, CAT is known to be inhibited by Ag^+^, cadmium(II) (Cd^2+^), chromium(VI) (Cr^6+^), copper(II) (Cu^2+^), and zinc(II) (Zn^2+^) in five tissues studied [83]. Titanium dioxide NPs have been shown to induce conformational and functional changes in both CAT and SOD, increasing the alpha helical content of these proteins and exposing more hydrophobic regions, as determined by ultraviolet-visible (UV-vis) and CD spectroscopies [77]. The majority of studies that exist on the effects of nanoparticles on CAT utilize nanoparticles to simulate the catalytic activities of both CAT and SOD (see reviews by He et al. (2014) [84] and Singh (2019) [85]). Interestingly, CAT encapsulated in nanoporous silica nanoparticles (CAT-Si NPs) has been used to generate oxygen to relieve hypoxia in tissues and potentially sensitize tumors against radiation therapy [86].

The superoxide dismutases are a family of ubiquitous, metal-binding antioxidant enzymes which catalyze the dismutation (or partitioning) of the superoxide (O_2_^−^) radical into ordinary molecular oxygen and H_2_O_2_. Three forms of the enzyme exist: the eukaryotic cytosolic form is copper-zinc SOD (Cu-Zn-SOD), the eukaryotic mitochondria/chloroplast form is either manganese SOD (Mn-SOD) or iron SOD (Fe-SOD), and a prokaryotic form is nickel SOD (Ni-SOD). The interaction of nanoparticles with the SODs (as well as CAT) has been studied to a limited extent. One study on the interaction of Ag NPs with CAT and SOD showed that interaction of Ag NPs with CAT caused significant conformational changes in the enzyme, resulting in loss of CAT catalytic activity, but had minimal effects on SOD shape and activity [69]. In this study, CAT was able to promote Ag NP dissolution, but the released Ag ions did not have an effect on its heme cofactor. Unlike CAT, Ag NPs did not have an effect on the SOD metal cofactors, as determined by UV-vis spectroscopy, fluorescence, and CD [69].

The GPxs are a family of enzymes with peroxidase activities towards organic hydroperoxides and H_2_O_2_. These enzymes are selenium-dependent and selenium (Se) is bound to a unique cysteine (selenocystine) within the enzyme, which alternates between a reduced selenol (R-SeH) form and an oxidized selenenic acid (R-SeOH) form. The selenocystine-dependent system is also found in thioredoxin reductase, used to reduce TRX. In kidney, cadmium-telluride quantum dots (CdTe-QD) have been shown to bind directly to glutathione peroxidase 3 (GPx3) via Van der Waals’ forces and hydrogen bonding, resulting in structural changes with increasing contents of α-helix in the enzyme [72]. Cadmium-telluride quantum dots were found to interact with glutamate 136, phenylalanine 132, proline 130, and valine 129 in the active site of the enzyme. In birds, Cd was found to inhibit the activity of, specifically, the Se-dependent GPxs [73]. In rat pheochromocytoma (PC12) cells, the co-presence of non-toxic Se (5 μM) and toxic Cd (5 μM) increased cell viability, glutathione (GSH), and glutathione peroxidase 1 (GPx1) levels [87]. Very few studies focus on the Se-dependent enzymes and the effects of nanoparticles or nanoparticle-derived metals. Most studies focus on the creation of Se NPs as a method of delivery of Se with potential antioxidant/anti-tumorigenic activities [88]. Selenium NPs have been shown to increase the activity of the Se-dependent GPx enzymes in numerous studies [76,89,90]. Other Se-dependent enzymes, such as thioredoxin reductase, have been shown to be inhibited by Ag NPs [70]. The expression of selenoproteins, in general, is decreased by Ag NP exposure of human epidermal keratinocyte cells (HaCaT) and human lung adenocarcinoma cells (A549) [70].

The interaction of nanoparticles with sulfhydryl (SH) groups in proteins is, in general, due to the dissolution of metal ions from the nanoparticles which subsequently interact with the SH group of cysteine and sulfide (SCH_3_) group of methionine [91]. Oxidized heavy metals replace the hydrogen of the SH group and the methyl of the SCH_3_ group, thereby inhibiting the function of these thiol-dependent enzymes [92]. The metals bridge the gap between two sulfurs within the same protein (intramolecular bonding) or between two different proteins (intermolecular bonding). If the sulfurs are present on methionines, this involves the release of two methyl groups from these amino acid side chains. Cadmium has been shown to inhibit human thiol-dependent antioxidant proteins such as thioredoxin reductase, GR, and TRX in vitro by binding to cysteine residues in their active sites [93]. Metals binding to thiols may also inactivate GSH as a low molecular weight antioxidant, preventing it from oxidizing, being conjugated to hydroxylated toxins (through the action of the glutathione S-transferases (GSTs)), and forming mixed disulfides with proteins (under oxidizing conditions; known as glutathionylation).

Glutathione reductase functions as a dimeric disulfide oxidoreductase that utilizes an flavin adenine dinucleotide (FAD) prosthetic group and nicotinamide adenine dinucleotide phosphate (NADPH) to reduce one molar equivalent of oxidized glutathione (GSSG) to two molar equivalents of reduced GSH. Cysteines 58 and 63 of the human enzyme form the redox active catalytic SH groups that interact with the GSSG substrate [94]. Exposure to higher amounts of chromium compounds in humans can lead to the inhibition of erythrocyte GR (as well as CAT, GPx, thioredoxin reductase, and glucose-6-phosphate dehydrogenase), which in turn, lowers the capacity to reduce methemoglobin to Hb [95]. Exposure of juvenile carp (*Cyprinus carpio*) to zinc caused reductions in the activity of GR in gills, liver, and brain [96]. Human erythrocyte GR was inhibited competitively by lead(II) (Pb^2+^), mercury(II) (Hg^2+^), Cd^2+^, and iron(III) (Fe^3+^), and non-competitively by Cu^2+^ and Al^3+^ [78]. These metals are commonly found in NPs and will dissociate as free ions as the NP breaks down, potentially inhibiting GR and affecting the GSH/GSSG ratios in cells.

The PRDXs, similar to the GPxs, are thiol-specific peroxidases that catalyze the reduction of H_2_O_2_ and organic hydroperoxides to water and alcohols. They play an important role in, not only preventing oxidative stress, but also as sensors of H_2_O_2_-mediated intracellular signaling events. A proteomics study undertaken in human hepatocytes and human hepatoma HepG2 cells has revealed that Ag NPs and Ag^+^ binds to PRDXs, as well as GST, myosin, elongation factor 1, 60S ribosomal protein, and 40S ribosomal protein [71]. A second study exposing human lung epithelial cells and human monocyte-derived macrophages to engineered nanosized rutile/anatase titanium dioxide (r/aTiO_2_), silica-coated nanosized rutile titanium dioxide (rTiO_2_ silica-coated), alumina-coated nanosized rutile titanium dioxide (rTiO_2_ alumina-coated), nanosized multiwalled carbon nanotube (mwCNT), and nanosized anatase titanium dioxide (aTiO_2_), also showed PRDXs associating with the nanoparticles [75]. Copper-based nanoparticles induced an increase in the protein levels of the oxidized form of PRDX1 and the native form of PRDX6 in mouse macrophages [74]. The levels of other PRDXs, such as PRDX2 and the mitochondrial PRDX3, remained unchanged in this study. The implications of the association of PRDXs with various types of nanoparticles remain to be determined.

The TRX/glutaredoxin/protein disulfide isomerase (PDI) systems are a family of thioltransferase proteins that catalyze reversible disulfide exchange reactions with other proteins. They possess two redox-active cysteines within a short conserved active site sequence (cysteine–glycine–proline–cysteine). Thioredoxin-1 (TRX1) is cytoplasmic, while thioredoxin-2 (TRX2) is mitochondrial. They also are responsible for removing GSH from glutathione-protein mixed disulfides (deglutathionylation). Finally, these thioltransferase proteins play a role in the reversible S-nitrosylation of cysteine residues in target proteins, and thereby contribute to the response to intracellular nitric oxide (^●^NO). Exposure of H9 human embryonic stem cells and Jurkat cells to Cd has been shown to inactivate thioltransferase activity, thereby inhibiting the intracellular reduction of protein-glutathionyl-mixed disulfides and initiating apoptosis [93]. A study by Hansen et al. (2006) [97] showed that metals such as copper, iron, and nickel showed significant oxidation of GSH but relatively little oxidation of either TRX1 or TRX2, whereas metals such as arsenic, cadmium, and mercury showed little oxidation of GSH but significantly oxidized both TRX1 and TRX2. The findings from this study show that metals have differential oxidative effects on the major thiol antioxidant systems and that activation of apoptosis may be associated with metal ions that oxidize TRX. Protein disulfide isomerase is associated with the response to endoplasmic reticulum (ER) stress, as well as oxidative stress, and studies have shown that the messenger ribonucleic acid (mRNA) of protein disulfide isomerase-3 (PDI-3) is upregulated in mouse liver in response to ZnO NP exposure [79].

## 3. Nanoparticles and the Immune Response

Nanoparticles tend to elicit an immune response in biological systems resulting in the production of inflammation and oxidative stress [98]. Foreign particles or pathogens that invade the body are initially combated by a nonspecific immune response regulated by the innate immune system as the first line of defence. Phagocytes such as macrophages, neutrophils, monocytes, dendritic cells, and mast cells release cytokines and work to engulf and destroy foreign bodies. The phagocytes are antigen-presenting cells, and display antigens from the foreign bodies for the T-lymphocytes (T-cells) and B-lymphocytes (B-cells) in the adaptive immune system to recognize, degrade, and store in their immunological memory [99,100]. Macrophages are crucial to the immune system as they are generally the first to detect and defend against foreign materials or pathogens by engulfing them and are likely the first immune cells to interact with NPs in the body [101]. Macrophages tend to interact well with positively charged nanoparticles due to the negatively charged sialic acid on their surface, generally resulting in more inflammation and toxicity being produced by positively charged NPs than negatively charged or neutral NPs [99,100,102]. Table 3 shows the effects that various NPs have on the immune response in various in vitro and in vivo models.

Nanoparticle exposure can influence the proliferation of the various immune cells. Exposure of female mice to CuO NPs by continuous inhalation for up to 93 days mostly affected the spleen cell populations of macrophages, neutrophils, eosinophils, and antigen-presenting cells in the innate immune system. It was observed that the effect on proliferation and cytokine production by the T-cells and B-cells in the adaptive immune system changed over time, with an increase by day 3 that decreased by day 14 and then normalized [109]. Gold NPs were found to have a low risk for producing a pulmonary immune response unless there was already a pre-existing allergy to Au. Female C57BL/6J mice treated with Au NPs by oropharyngeal aspiration, experienced no change in lymphocyte cell population in the mediastinal lymph nodes in the mice not sensitive to Au. However, an increased T-cell population was observed in mice pre-sensitized to Au and treated multiple times with Au NPs [112]. Amino-functionalized carbon nanotubes decreased the cell populations of monocytes and macrophages in peripheral blood mononuclear cells (PBMCs), but interestingly did not have this effect in vivo with C57Bl/6 mice [113]. An antibacterial Ag NP infused hydrogel (made from thiolated chitosan and maleic acid-grafted dextran) has been developed and found to stimulate the immune system with increased numbers of T-cells and macrophages, and significantly decreased wound healing time in a diabetic rat model [104]. Silver NPs have also been found to induce B-cell proliferation in male Naval Medical Research Institute (NMRI) mice [105]. In the immune system, cytotoxic T-cells are generally antigen specific, and once activated through recognition of their specific antigen on antigen presenting cells undergo clonal expansion, travel throughout the body, and work to clear the triggering virus, bacteria, or cancer. Interestingly, treatment of cytotoxic T-cells with cerium (Ce) NPs activated them and increased their cytotoxic activity. Cerium NPs have antioxidant properties and decreased ROS production in the activated cytotoxic T cells, which induced nuclear factor kappa B (NF-κB) signaling, leading to increased cytokine production of interleukin (IL)-2 and tumor necrosis factor alpha (TNF-α), and increased production of the effector molecules granzyme B and perforin, all resulting in increased killing activity [108].

### 3.1. Nanoparticle Effects on Immune Cell Receptors

To recognize foreign bodies, phagocytes have pattern recognition receptors (PRRs) that recognize ligands by their pathogen-associated molecular patterns (PAMPs). Phagocytes also have receptors for damage-associated molecular patterns (DAMPs) that sense the level of surrounding tissue damage. Toll-like receptors (TLRs) recognize PAMPs, with specific TLRs recognizing specific ligands [99]. Binding to PAMPs stimulates defence mechanisms and results in the secretion of pro-inflammatory mediators that initiates the inflammatory response, increases antigen presentation, and encourages immune cell phagocytosis and cytolytic activity [117,118]. Inflammatory mediators secreted by immune cells include cytokines, chemokines, histamine, proteases, prostaglandins, leukotrienes, and serglycin proteoglycans [119]. Pro-inflammatory cytokines include IL-1β, IL-1α, IL-6, IL-8, IL-12, TNF-α, interferon gamma (IFN-γ), and monocyte chemoattractant protein-1 (MCP-1). The anti-inflammatory cytokines are IL-4, IL-10, IL-11, and transforming growth factor-beta (TGF-β) [119,120]. Various NPs have been found to activate TLRs and induce inflammation [121].

Various nanoparticles are able to attenuate situations of induced immune response and inflammation. Lipopolysaccharide (LPS) is a major component of the outer membrane of Gram-negative bacteria. It is a potent agonist for TLR-4, induces sepsis and toxic shock syndrome, and is commonly used experimentally to induce an inflammatory and immune response [117,122,123]. Superparamagnetic iron oxide NPs have an anti-inflammatory effect and attenuated the immune and inflammatory response induced by LPS in RAW 264.7 murine macrophages. The treatment decreased the expression levels of TLR-4 and nitric oxide synthase (NOS) and reduced the release of inflammatory cytokines IL-6 and TNF-α [117]. Gold NPs and Ag NPs have been found to decrease immune cell proliferation stimulated by LPS (a B-cell mitogen) and Concanavalin A (a T-cell mitogen) in murine splenic lymphocytes. These NPs also decreased the immune cell proliferation that was stimulated by phytohemagglutinin (a T-cell mitogen) and pokeweed mitogen (a T-cell and B-cell mitogen) in human blood lymphocytes. Interestingly, unstimulated lymphocytes were not affected by exposure to Au and Ag NPs [116]. Rats without an induced immune response treated with Ag NPs for 28 consecutive days resulted in accumulation in the spleen, liver, and lymph nodes; however, no accompanying inflammation was observed. The spleen had increased T-cell, B-cell, and natural killer cell proliferation; decreased natural killer cell activity; and decreased production of the pro-inflammatory cytokine IFN-γ [111]. In a follow-on study with an immune response induced by keyhole limpet hemocyanin (KLH), exposure to Ag NPs decreased the induced level of KLH immunoglobulin G (IgG), indicating a beneficial calming of the immune response [115].

In situations without an induced immune response, certain NPs trigger an immune response while others decrease it. For example, treatment of mouse macrophages with superparamagnetic iron oxide NPs activated TLR-4 receptors, increased the expression of inflammatory cytokines, triggered the translocation of the antioxidant response transcription factor nuclear factor erythroid 2-related factor 2 (Nrf2) to the nucleus, stimulated autophagy, and increased the gene expression of membrane type I class A scavenger receptor (SR-AI) (which has been found to recognize superparamagnetic iron oxide NPs in macrophages) [110]. Treatment of human acute myeloid leukemia Tohoku Hospital Pediatrics-1 (THP-1) cells with boehmite (AlOOH) NPs, CuO NPs, TiO_2_ NPs, and ZnO NPs increased TLR-4 gene expression; AlOOH NPs, CuO NPs, TiO_2_ NPs, ZnO NPs, Ag_2_O NPs, and iron oxide NPs activated TLR-6 [106]. Silver NPs were internalized by RAW 264.7 cells and induced the expression of TLR-3 on the macrophage surface with no change in TLR-4 expression [105]. Aluminum NPs accumulated mainly in the liver, with some being deposited in the kidney, brain, spleen, lungs, testis, and heart of male ICR mice. The spleen, thymus, and bone marrow are immune system organs, and aluminum NP treatment resulted in oxidative damage, neutrophil dysfunction, and increased expression of TNF-α, IFN-γ, IL-1α, IL-1β, IL-2, IL-6, and IL-10 [107]. Interestingly, graphene and multi-wall carbon nanotubes decreased the gene expression of TLR-5 and decreased the production of pro-inflammatory cytokines IL-1β and IL-6 in mouse macrophage J774 cells [114].

### 3.2. Nanoparticle Effects on Oxidative Stress and the Immune Response

Oxidative stress in biological systems is caused by an imbalance between the levels of antioxidants and the levels of ROS or reactive nitrogen species (RNS), which are forms of oxygen or nitrogen with an unpaired electron in their outer electron orbital. Active phagocytic immune cells produce ROS and RNS as anti-microbial and killer molecules to combat foreign agents by triggering enzymes such as inducible nitric oxide synthase (iNOS) to produce ^●^NO and NADPH oxidase to produce O_2_^−^ [122,123].

Silica NPs have been found to produce ROS and induce apoptosis in human lymphocyte cells collected from blood samples [124]. Silver NPs increased the level of ^●^NO and triggered the production of GSH in human promyelocytic leukemia cells (HL-60) that had been differentiated to macrophage-like cells [125]. Graphene oxide NPs induced the production of ROS and lipid peroxidation, and decreased the level of the cellular antioxidants GSH, GPx, SOD, and CAT in human acute myeloid leukemia THP-1 cells. Additionally, an increase in pro-inflammatory cytokines and chemokines IL-1β, TNF-α, IL-6, IL-8, and MCP-1 indicated that inflammation was induced [126].

### 3.3. Summary

Depending on their properties, some NPs (CuO, Ag, ALOOH, TiO_2_, ZnO, Fe_3_O_4_, and aluminum) tend to induce an immune response in biological systems, while other NPs with antioxidant properties (Au, Ce, graphene, carbon nanotubes) work to decrease it. Fascinatingly, several NPs (Ag and superparamagnetic iron oxide) that are reported to induce an immune response and inflammation when examined on their own in experimental models have the ability to attenuate situations of induced immune response and inflammation. Gold NPs have beneficial effects and are able to calm an induced immune response while not inducing an immune response themselves (unless a pre-existing gold allergy is present).

## 4. Inflammatory and Anti-Inflammatory Effects of Nanoparticles

Inflammation is an immune response to tissue injury caused by various factors such as pathogens, toxic substances, and cell damage. The immune system triggers inflammation to repair tissue damage and defend against foreign substances in the body. Acute inflammation is integral for healing; however, long-term chronic inflammation or an excess production of pro-inflammatory cytokines may result in tissue damage, organ failure, various diseases, and potentially death. In fact, chronic inflammation has been known to be involved in a variety of diseases such as cancer, cardiovascular disease, diabetes, hypertension, and atherosclerosis [119,127]. There are many studies that have shown that exposure to various NPs including ZnO NPs, Al_2_O_3_ NPs, TiO_2_ NPs, CeO_2_ NPs, Ag NPs, CuO NPs, Ni NPs, SiO_2_ NPs, iron oxide NPs, and carbon nanotubes induces inflammation in various in vitro and in vivo studies [24,25,99,120,128,129,130,131,132,133,134,135,136,137,138]. Various properties of the NPs, such as size, charge, surface reactivity, and surface coating, can alter the inflammatory response in biological systems and it has been generally found that smaller NPs tend to induce more inflammation than larger NPs [136,137,139,140].

The most common signaling pathways of the immune system are the NF-κB, MAPK, and the Janus kinase (JAK); signal transducer; and activator of transcription (STAT) pathways. The MAPK and NF-κB signaling pathways recruit immune cells which are responsible for producing inflammatory cytokines at the site of injury [119]. Released cytokines and other inflammatory mediators function as soluble intercellular messengers to communicate with and recruit other immune cells and thus are integral in regulating inflammation and the immune response [119,141,142]. Many studies have found that various NPs induce an increase in both gene and/or protein expression of pro-inflammatory cytokines [25,98,117,131,134,136,137,143,144,145,146,147,148], while other NPs have been found to be anti-inflammatory [136,149,150,151,152,153,154,155,156]. The transcription factor NF-κB has a central role in regulating the production of pro-inflammatory cytokines such as TNF-α, IL-8, IL-2, and IL-6, while MAPKs are involved in regulating the production of pro-inflammatory cytokines such as TNF-α, IL-1, IL-6, IL-8, and MCP-1 [101,157]. Nanoparticles such as ZnO NPs, Al_2_O_3_ NPs, TiO_2_ NPs, CeO_2_ NPs, Ag NPs, CuO NPs, and quantum dots affect the NF-κB and/or MAPK pathways either by activation of one of the pathways or by affecting the proteins involved in the pathways [98,101,133,137,158,159,160,161,162]. Calcium signaling plays a role in the activation of various pathways, including the MAPK and NF-κB pathways, and may result in the production of pro-inflammatory cytokines [163]. Thus, NPs that alter calcium homeostasis may also affect the signaling pathways involved in the inflammatory response.

The inflammasome is a protein complex that plays an important role in the inflammatory response. Inflammasome activation results in the release of pro-inflammatory cytokines IL-1β and IL-18 [128,130,157,164]. Currently, there are several known inflammasomes including nucleotide-binding oligomerization domain, leucine rich repeat and pyrin domain containing 1 (NLRP1), NLRP3, absent in melanoma 2 (AIM2), and NLR family CARD domain containing 4 (NLRC4), with NLRP3 being the most widely studied inflammasome [132,165]. Various NPs are known to interact with the NLRP3 inflammasome, especially in experimental situations where the NP exposure is through inhalation [128,130,134,166].

Nanoparticles tend to induce the production of ROS [98,101,131,136,163]. Oxidative stress may induce inflammation through affecting redox sensitive pathways and inducing a release of pro-inflammatory cytokines [142]. Inflammation may also induce oxidative stress [157]. An alternative pathway to induce inflammation not involving oxidative stress has been identified for NPs that can bind to fibrinogen and induce unfolding. This can trigger the activation of the macrophage-1 antigen (Mac-1) receptor leading to activation of NF-κB signaling and inflammation. This was demonstrated with negatively charged poly(acrylic acid) conjugated Au NPs in THP-1 cells [167].

### 4.1. Effects of Silver Nanoparticles on Inflammation

Silver NPs have been found to induce anti-inflammatory effects in some experimental models, while others have found inflammatory effects. Treatment of LPS-induced RAW 264.7 macrophages with Ag NPs decreased the LPS-induced expression of inflammatory mediators including iNOS, ^●^NO, prostaglandin E2, and cyclooxygenase-2 (COX-2), and inhibited the LPS-induced activation of the MAPK and NF-κB signaling pathways [168]. A study looking at lung epithelial cells found that Ag NPs interacted with the cells and decreased the expression of TNF receptor 1, resulting in a decrease of TNF-α signaling. This leads to a potential decrease in production of pro-inflammatory cytokines [169]. Silver NPs inhibited the production of pro-inflammatory cytokines in male Swiss mice [165] and RAW 264.7 macrophages [170]. The MAPK, interleukin signaling, and TGF-β pathways changed in human glioblastoma cells in response to Ag NP exposure. This was seen through decreases in gene expression, indicating that Ag NPs may have an anti-inflammatory effect [171]. The anti-inflammatory effect of Ag NPs has also been reported in immortalized embryonic mouse microglia cells [153]. Silver NPs decreased the NLRP3 inflammasome levels in primary bone marrow-derived macrophages (BMDMs) isolated from male C57/BL6 mice [172].

Other experimental models have found Ag NPs to have inflammatory effects. Silver NPs induced a release of pro-inflammatory mediators in various neural cells including microglia, astrocytes, and neurons [173,174,175]. Silver NPs have also been found to activate the NLRP3 inflammasome through the induction of ROS, ER stress, and cell membrane damage [166,176,177,178,179,180]. In RAW264.7 cells, Ag NPs induced activation of the NF-κB pathway, resulting in upregulation of pro-inflammatory genes [181]. Treatment of NIH3T3 murine fibroblast cells with Ag NPs increased oxidative stress, which then activated c-Jun N-terminal Kinase (JNK), a protein involved in the MAPK pathway [182]. Exposure to Ag NPs induced oxidation of arachidonic acid to form 5-Hydroxyicosatetraenoic acid (5-HETE), a pro-inflammatory chemoattractant, in THP-1 cells [183]. A reduction in production of ^●^NO was observed after exposure to Ag NPs in PBMCs, THP-1 cells, and in murine macrophages [184,185].

### 4.2. Pro-Inflammatory Effects of Titanium Dioxide Nanoparticles

Titanium dioxide NPs have been found to trigger inflammation in many studies. Various in vitro and in vivo studies have found that exposure to TiO_2_ NPs results in increased levels of pro-inflammatory cytokines and proteins [136,186,187,188,189,190,191]. It has been found that as the concentration of TiO_2_ NPs increases, the M1 phenotype of macrophages (distinguished for releasing pro-inflammatory cytokines, ROS, ^●^NO, and antigen presentation) became more active than the M2 phenotype of macrophages (which secrete anti-inflammatory cytokines, regulates inflammation, and promotes healing) in male Wistar rats [131]. Glial cells, which include microglia and astrocytes, play a crucial role in mediating inflammation in the nervous system [192]. Microglia cells growing on a TiO_2_ nanostructured surface displayed conversion to the M1 inflammatory phenotype. Interestingly, this was not observed in astrocytes grown on the same surface [173]. Isolated human macrophages treated with TiO_2_ NPs induced the secretion of IL-6, while a higher treatment level additionally induced the secretion of IL-1β and IL-10 [188]. Interestingly, a gel electrophoresis study examining the interaction of TiO_2_ NPs with pro-inflammatory cytokine and chemokines IF-γ, IL-1β, TNFα, IL-6, and C-X-C motif chemokine ligand 8 (CXCL8) found that TiO_2_ NPs selectively bind to the pro-inflammatory mediators CXCL8 and IFN-γ in a dose dependent fashion. The binding interaction between TiO_2_ NPs and CXVL8 was found to be stronger than between TiO_2_ NPs and IFN-γ. Additionally, the binding of the TiO_2_ NPs on the CXCL8 hampered the chemoattractant function of CXCL8 in the recruitment of neutrophils [45].

Oxidative stress due to TiO_2_ NP exposure is thought to trigger an inflammatory response in biological systems [163]. It has been found that TiO_2_ NPs triggered ROS generation and activated human dendritic cells in a dose-dependent manner, resulting in a pro-inflammatory state [136]. A study that examined the cellular uptake of TiO_2_ NPs in HepG2 and human chronic myeloid leukemia K562 cells found that TiO_2_ NPs bind to TLR3, TLR4, and TLR7 in the cell membrane, resulting in the uptake of TiO_2_ NPs and the activation of the NF-κB pathway [193]. Titanium dioxide NPs activated human umbilical vein endothelial cells (HUVECs), which then increased the expression of adhesion molecules and led to the recruitment and increased adhesion of monocytes to the endothelial cells. It also increased ROS and ^●^NO production and activated the NF-κB pathway [194]. Activation of NF-κB, resulting in upregulation of pro-inflammatory genes, has been seen in RAW264.7 cells treated with TiO_2_ NPs [181]. The mediators, extracellular signal-regulated kinase 1 and 2 (ERK1/2) and p38, in the NF-κB pathway were activated by TiO_2_ NPs in human dermal fibroblasts [195]. Increased gene expression of TLR, IL-1β, and MAPK signaling genes were found in CRL-2404 human epidermal keratinocytes treated with TiO_2_ NPs [161]. This indicates that TiO_2_ NPs may also play a role in MAPK activation to induce an inflammatory response.

It has been found that ROS, potassium efflux, lysosomal damage, and cathepsin B release are all involved in inflammasome activation by NPs [196]. Titanium dioxide NPs have been found to activate the NLRP3 inflammasome in human dendritic cells [136]. Altered levels of adenosine triphosphate (ATP), adenosine diphosphate (ADP), and adenosine by TiO_2_ NPs resulted in the activation of the NLRP3 inflammasome in THP-1 cells [186]. Activation of NLRP inflammasomes by TiO_2_ NPs, particularly the NLRP3 inflammasome, has also been found [186,189,190,197].

When the protein corona around TiO_2_ NPs in the cell culture media was examined, it was found to contain many post-translationally modified proteins that were phosphorylated or glycosylated, with albumin being one of the main components. It is thought that this protein corona around the TiO_2_ NPs would have a role in interacting with the macrophage surface receptors to induce the inflammatory response [188].

### 4.3. Effects of Gold Nanoparticles on Inflammation

Gold NPs have been found to have mainly anti-inflammatory effects in biological systems. Treatment with Au NPs prevented neuroinflammation in mice with induced Alzheimer’s disease [198]. Gold NP treatment of rats with acute carrageenan-induced inflammation inhibited the carrageenan-induced production of pro-inflammatory cytokines IL-1β and TNF-α, lipid peroxidation, and thiol group oxidation, and induced an increase in anti-inflammatory cytokine IL-10 [149]. Decreased levels of NF-κB and IL-1β were observed in mice exposed to Au NPs, preventing inflammation in the brain [199]. Additionally, anti-inflammatory effects of Au NPs were found in LPS-induced RAW 264.7 mouse macrophage cells. The Au NPs suppressed the production of pro-inflammatory cytokines and inhibited the activation of the JAK-STAT pathway [150]. Gold NPs had anti-inflammatory properties in LPS-induced mouse primary microglia as indicated by decreased production of inflammatory cytokines and mediators, and downregulation of the LPS-induced NF-κB, MAPK, and JAK-STAT signaling pathways [160]. Similarly, Au NP treatment in LPS-induced RAW 264.7 macrophages decreased the LPS-induced expression of inflammatory mediators including iNOS, ^●^NO, prostaglandin E2, and COX-2; and inhibited the LPS-induced activation of the MAPK and NF-κB signaling pathways [168]. However, another study found that Au NPs had no effect on the level of pro-inflammatory mediators TNF-α, IL-6, or ^●^NO in RAW 264.7 murine macrophages with or without LPS-induced inflammation. Additionally, the Au NPs did not induce ROS [151]. Gold NPs were found to decrease cytokine expression and reduce the gene expression of chemokine receptors in THP-1 cells, as well as to slow the recruitment of immune cells to a wound in male C57BL/6 mice that had been irradiated to destroy their own immune cells and then transplanted with Au NP treated macrophages [152].

Increased production of pro-inflammatory cytokines IL-1β, IL-1α, and IL-8 has been found with non-toxic Au NP treatment in primary neonatal human epidermal keratinocytes; however, the presence of a human plasma protein corona on the Au NPs decreased the NP uptake and no inflammatory response was observed [132].

### 4.4. Pro-Inflammatory Effects of Silicon Dioxide Nanoparticles

Silicon dioxide NPs have been found to induce inflammation in several experimental models. It has been found that SiO_2_ NPs induced the gene expression of inflammatory cytokines TNF-α, IL-6, and IL-8, and induced inflammation through elevated thioredoxin-interacting protein (TXNIP), MAPK, and activator protein-1 (AP-1) signaling in NCI-H292 human airway epithelial cells and female BALB/c mice. Interestingly, silibinin, a polyphenolic flavonoid from milk thistle which has antioxidant and anti-inflammatory properties, was able to attenuate the SiO_2_ NPs induced inflammation [133]. Amorphous SiO_2_ NPs produced ROS in vascular endothelial cells, resulting in activation of MAPK signaling, autophagy, and apoptosis [200]. When the effects of SiO_2_ NPs on the expression of 84 genes involved in MAPK signaling in Huh7 human hepatoma cells were examined, it was found that there was a strong induction of MAPK target genes. Exposure to SiO_2_ NPs also increased the expression of pro-inflammatory cytokines in TNF-α and IL-8 and induced ER stress [158].

Treatment of ovalbumin-induced asthmatic mice with SiO_2_ NPs activated the NLRP3 inflammasome, increased the number of inflammatory immune cells, inflammatory cytokine production (IL-1β, IL-6, and TNF-α), inflammation in the airway, mucus secretion, and increased protein expression of TXNIP [130]. Similarly, nontoxic treatment of human airway epithelial NCI-H292 cells with SiO_2_ NPs resulted in activation of the NLRP3 inflammasome and an increase in gene expression of pro-inflammatory cytokines IL-1β, IL-6, and TNF-α. This study proposed that SiO_2_ NPs induced activation of the NLRP3 inflammasome indirectly through the production of ROS, which then oxidized TRX into TXNIP, and led to activation of the NLRP3 inflammasome [130]. Activation of the NLRP3 inflammasome by SiO_2_ NPs has also been found in murine dendritic cells [197].

### 4.5. Effects of Various Carbon Nanoparticles on Inflammation

It has been found that graphene-based NPs are able to induce inflammation and/or expression of pro-inflammatory cytokines in various in vitro and in vivo models [201,202,203,204,205,206]. Carbon black NPs have been found to interact with and adsorb onto the pro-inflammatory cytokines IL-8 and TNF-α. Larger 260 nm carbon black NPs adsorbed onto TNF-α but did not hinder its function, shown by the ability of TNF-α to induce the expression of intercellular adhesion molecule 1 (ICAM-1) in A549 cells. However, smaller 14 nm carbon black NPs absorbed onto TNF-α more than the larger 260 nm carbon black NPs and appeared to inhibit the function of TNF-α since it could no longer induce ICAM-1. It was also found that exposure of HL-60 cells to carbon black NPs resulted in increased intracellular calcium levels, which has the potential to activate inflammatory signaling [207].

Graphene oxide NPs have the potential to affect immune cell function such as T lymphocytes, macrophages, and THP-1 cells [201,208,209,210,211]. Graphene oxide NPs have been found to bind to TLRs and activate the NF-κB pathway in primary mouse macrophages [212]. Capillary walls in the alveoli in male C57BL/6 mice were disrupted by graphene oxide NPs, resulting in infiltration of immune cells into the lungs, and induction of pro-inflammatory cytokine production [202].

In contrast, gadofullerene NPs exhibited anti-inflammatory effects in diabetic mice, and decreased the gene expression of various pro-inflammatory cytokines including NF-κB, TNF-α, IL-6, and IL-1β [154].

### 4.6. Anti-Inflammatory Effects of Cerium Nanoparticles

Cerium NPs have potent antioxidant properties due to their ability to cycle between the Ce^3+^ and Ce^4+^ oxidation states and mimic the activity of SOD and CAT in biological systems [108]. Oxidative stress and inflammation are closely linked, and Ce NPs are also anti-inflammatory. Human dendritic cells treated with CeO_2_ NPs increased production of the anti-inflammatory cytokine IL-10 and did not activate NLRP3 inflammasome signaling [136].

Cerium dioxide NPs have anti-inflammatory, antioxidant, and antimicrobial properties, and are being explored for the treatment of sepsis with promising results. Male Sprague Dawley rats with cecal inoculum-induced sepsis treated with CeO_2_ NPs intravenously at 0.5 mg/kg body-weight decreased the sepsis-induced inflammation, diaphragm contractile dysfunction, and iNOS gene and protein expression [36]. Male Sprague Dawley rats with LPS-induced sepsis treated with intravenous CeO_2_ NPs at 0.5 mg/kg body-weight also showed increased survival, decreased liver damage, and decreased inflammation. The mechanisms for these positive results were examined using Kupffer cell macrophages extracted from the liver of the rats, and CeO_2_ NP treatment decreased LPS-induced inflammatory cytokines and ROS [213].

### 4.7. Pro-Inflammatory Effects of Zinc Oxide Nanoparticles

Zinc oxide NPs have demonstrated pro-inflammatory effects in several in vitro studies. Human acute myeloid leukemia THP-1 cells treated with ZnO NPs increased the production of ROS and ^●^NO, which in turn activated the NF-κB and MAPK pathways, resulting in increased production of pro-inflammatory cytokines [137]. Macrophage cells from female Balb/c mice treated with ZnO NPs showed activation of the MAPK and NF-κB pathways [101]. Primary dendritic cells isolated from C57BL6/J mouse bone marrow were activated by treatment with ZnO NPs and increased the production of ROS and pro-inflammatory cytokines IL-6 and TNF-α [214].

### 4.8. Anti-Inflammatory Effects of Selenium Nanoparticles

Selenium has been found to have neuroprotective effects and anti-inflammatory effects [215]. Selenium NPs have been found to have beneficial and anti-inflammatory effects in biological systems. In a type 2 diabetes rat model, Se NPs decreased the expression of pro-inflammatory cytokines [155]. Selenium NP treatment of RAW 264.7 macrophages decreased the gene expression of pro-inflammatory cytokines IL-1 and TNF-α, increased the gene expression of the anti-inflammatory cytokine IL-10, and inhibited the NF-κB and MAPK pathways [156]. A study looking at the effects of SeNPs in male albino rats showed that SeNPs had inhibited DNA damage, neuroprotective effects, and protective effects to the kidneys [216].

A decrease in pro-inflammatory cytokine expression was found in an Alzheimer’s mouse model after treatment with Cd-Se quantum dots. Neuroinflammation decreased and learning and memory ability were improved [217].

### 4.9. Summary

It is important to examine whether NPs in biological systems have pro-inflammatory or anti-inflammatory effects, whether any pro-inflammatory effects are acute or chronic, and the effects of dose and exposure time. Short-term pro-inflammatory effects may be beneficial and speed up the healing process, while long-term pro-inflammatory effects can be detrimental and result in the development of various diseases. Anti-inflammatory NP effects are beneficial in diffusing a situation of chronic inflammation. Several NPs such as Au NPs, Ag NPs, Se NPs, Ce NPs, and gadofullerene NPs have been found to have anti-inflammatory effects.

Various NPs have been found to induce inflammation both in vitro and in vivo through different mechanisms including production of ROS, TLR signaling, the activation of inflammatory pathways, inflammasome activation, and induction of pro-inflammatory cytokines. A summary of the effects of various nanoparticles on inflammation in a variety of in vivo and in vitro models is shown in Table 4.

## 5. Effects of Nanoparticles on Mitochondrial Function

Mitochondria are critical to maintaining cellular function and homeostasis in many ways, including energy production via oxidative phosphorylation, body heat production, calcium storage and signaling, cell signaling, and apoptosis [220]. These fascinating membrane-bound organelles form dynamic reticular networks throughout the cell, undergo mitochondrial fission and fusion according to the cellular conditions and energy requirements, congregate where they are needed in the cell, and communicate with the other cellular organelles [221]. Cells in organs with higher energy requirements such as the heart, brain, liver, and skeletal muscle have large numbers of mitochondria, with the egg cell containing the most due to the high need for energy during mitosis. Mitochondrial fusion encourages increased energy production and cell growth [222]. Mitochondria are the master regulators of cell danger signaling and can sense various cellular stressors such as viral infection or nutrient deprivation and respond to the danger by remodeling the mitochondrial network and releasing signals to the cell or whole body to induce adaptive responses [223]. Under stress, mitochondria can be fragmented through too much fission and suffer mitochondrial dysfunction [224]. The function of damaged mitochondria may be restored by fusion to healthy mitochondria. If they are too damaged, they are degraded by mitophagy, which prevents the damaged mitochondria from releasing pro-apoptotic proteins and triggering intrinsic apoptosis. If cellular stress and damage is unrecoverable and cellular homeostasis cannot be regained, the mitochondria can determine the fate of the cell and mediate cell death through intrinsic apoptosis [134,225]. Many proteins control whether or not apoptosis occurs. The B-cell lymphoma-2 (Bcl-2) protein family is a large family with many functions and contains both anti-apoptotic proteins and pro-apoptotic proteins. Anti-apoptotic Bcl-2 proteins include anti-apoptotic Bcl-2 and Bcl-XL, which inhibit the release of Cyt c. Pro-apoptotic Bcl-2 proteins include Bcl-2 associated X protein (Bax), which translocates from the cytosol to the mitochondria during apoptosis, and Bcl-2 homologous antagonist/killer (Bak), which resides in the mitochondria [225]. Tumor suppressor protein p53 can activate the transcription of pro-apoptotic proteins and bind to and inhibit anti-apoptotic Bcl-2 proteins to allow the release of Cyt c [226]. Cytochrome c is normally part of the electron transport chain (ETC) on the inner mitochondrial membrane in healthy cells and passes electrons from complex III to complex IV. Mitochondrial outer membrane permeabilization is the point of no return in apoptotic and necrotic pathways, with this releasing Cyt c among many other mitochondrial proteins into the cytosol. Released Cyt c binds to apoptotic protease-activating factor-1 (Apaf-1). Binding of ATP or deoxyadenosine triphosphate (dATP) to this complex induces oligomerization to occur, forming an apoptosome complex. The apoptosome activates initiator caspase-9, which activates executioner caspase-3 and -7, resulting in a caspase signaling cascade, chromatin fragmentation, and externalization of phosphatidylserine which signals the phagocytes for removal by phagocytosis [227]. Other proteins released from the mitochondria, such as Second Mitochondria-Derived Activator of Caspase (Smac) and Omi, bind to Inhibitor of Apoptosis Proteins (IAPs), stopping them from inhibiting the activation of the caspases [223,225,228].

Various aspects of mitochondrial mediated intrinsic apoptosis have been observed many times in vitro with a wide range of NPs, such as with A-357 human skin melanoma cells treated with Cu NPs [229], HepG2 cells treated with Ag NPs [230], HCT116 human colon carcinoma cells treated with Ag NPs [231], MCF-7 breast adenocarcinoma cells treated with <200 nm *Rubus fairholmianus* extract conjugated Ag NPs [232], human neuroblastoma SH-SY5Y cells treated with SiO_2_ NPs [21], PC12 cells treated with superparamagnetic iron oxide NPs [233], HepG2 cells treated with cadmium sulphide quantum dots [234], human oral cancer cells treated with phloretin loaded chitosan NPs [235], human chronic myeloid leukemia K562 cells treated with N-Succinyl-chitosan NPs [236], A549 cells treated with nanoplastic polystyrene NPs [52], HT-29 colon cancer cells treated with Au NPs stabilized with polyphenols from *Abutilon indicum* leaf extract [237], MCF-7 cells treated with photothermal therapy using nuclear-targeting Au nanostars (Au NSs) with a core diameter of 20 ± 2 nm and arm length of 36 ± 3 nm [238], and murine embryonic fibroblasts (MEFs) transformed with the simian virus 40 (SV40) virus treated with photothermal therapy using laser-irradiated Au nanoprisms [239]. Mitochondrial mediated apoptosis has also been seen in human aortic endothelial cells (HAECs) treated with ZnO NPs, resulting in decreased mitochondrial membrane potential, Cyt c release, activated caspase-9 and caspase-3, and an increased Bax/Bcl-2 ratio. Interestingly, co-treatment of the ZnO NPs with the antioxidant alpha-lipoic acid was able to alleviate their detrimental effects [240]. Protective effects have also been demonstrated with selenium against Ag NP induced mitochondrial membrane depolarization, ROS production, and caspase-3 activation in murine hippocampal neuronal HT22 cells [241].

Mitochondrial mediated apoptosis has been observed in vivo as well, for example, with uterine damage in female Sprague-Dawley rats treated with Cu NPs at 3.12, 6.25, 12.5 mg/kg body-weight for 14 days by intraperitoneal injection [242]. Additionally, kidney damage was observed in Swiss albino male mice treated with Cu NPs by oral gavage for 3 days at 200, 413, and 600 mg/kg body-weight, with evidence of oxidative stress and both intrinsic and extrinsic apoptosis in the kidney tissue [243].

On the other hand, NPs made from derivatives of water-soluble fullerene have remarkable antioxidant properties, and NPs made from the bis-adduct malonic acid fullerene derivative, C_60_(C(COOH)_2_)_2_, have been found to protect human cervical carcinoma (HeLa) cells from stimulated decrease in mitochondrial membrane potential, Cyt c release, and mitochondrial meditated cell death [244].

### 5.1. Effects of Nanoparticles on Mitochondrial Fission and Fusion

Nanoparticle exposure commonly causes mitochondrial dysfunction in biological systems. Exposure of living organisms or isolated cells to NPs tends to result in increased mitochondrial fission and decreased mitochondrial fusion. In the mitochondrial fission pathway, Dynamin-related protein 1 (Drp1) becomes active once phosphorylated (p-Drp1) on serine 616 and binds to the mitochondrial fission protein 1 (Fis1), leading to mitochondrial fission, fragmentation, and potentially apoptosis. In mitochondrial fusion, the outer membranes first fuse together, with this process mediated by mitofusions 1 (Mfn1) and mitofusions 2 (Mfn2). The inner membranes then fuse together, mediated by optic atrophy 1 (Opa1).

Treatment of Sprague-Dawley rats with Ag NPs by a single intratracheal instillation was found to adversely affect the mitochondria in the lung tissue, and shifted the remodeling of the mitochondria towards increased fission and decreased fusion [224]. In this study, the protein expression of the fission proteins Drp1, p-Drp1, and Fis1 increased, while the protein expression of the fusion proteins Opa1 and Mfn2 decreased. Additionally, the mitochondria were swollen, Ag NPs were visible inside the mitochondria, and apoptosis was observed through increased protein expression of cleaved caspase-3. Interestingly, treatment of the rats with selenium in the form of sodium selenite after the Ag NP exposure increased detoxification in the rats and alleviated all of these detrimental effects [224].

Silver NPs have been found to increase expression of fission proteins, decrease expression of fusion proteins, and increase fragmentation in human induced pluripotent stem cells (iPSCs) [245] and HepG2 human liver carcinoma cells [229]. Additionally, in HepG2 cells, a decrease in the protein expression of peroxisome proliferator-activated receptor gamma coactivator 1-alpha (PGC-1α), a transcription factor involved in mitochondrial biogenesis, was observed [222]. Gold NPs have many applications in cancer treatment for drug delivery systems, photothermal therapy, and diagnostic bioimaging systems; and have the ability to inhibit tumor growth, decrease angiogenesis, and decrease metastasis [246]. When combined with TNF-related apoptosis-inducing ligand (TRAIL), Au NPs augmented the anti-tumor effects of TRAIL in female BALB/c mice subcutaneously injected with non-small-cell lung cancer (NSCLC) cells and increased mitochondrial fragmentation and fission in NSCLC cells through increasing the recruitment of Drp1 protein to the mitochondria [246]. Silica NPs have been found to detrimentally effect the mitochondria in HUVECs, as seen with the mitochondrial uptake of Si NPs, mitochondrial swelling, visible disturbance and destruction of the cristae, increased mitochondrial ROS, mitochondrial membrane potential collapse, and decreased ATP levels. They also changed the expression levels of fission and fusion markers, with increased gene or protein expression of Fis1, Drp1, and Mfn2, and decreased gene and protein expression of Opa1 [247]. Treatment of human pulmonary artery endothelial cells (HPAECs) with carbon black FW2 NPs resulted in increased intracellular and mitochondrial calcium levels, increased mitochondrial superoxide production, decreased mitochondrial membrane potential, increased mitochondrial swelling, decreased mitochondrial cristae, increased mitochondrial fragmentation, and increased mitochondrial mediated apoptosis with increased caspase-9 activity [49].

### 5.2. Effects of Nanoparticles on Energy Production, Oxidative Phosphorylation, and Glycolysis

Under normal conditions, healthy cells mainly rely on oxidative phosphorylation in the mitochondria for their energy production. In this process, a proton gradient is set up via the ETC which is used to produce energy through chemiosmosis. The proton gradient is set up across the inner mitochondrial membrane by complex I (reduced nicotinamide adenine dinucleotide (NADH)-coenzyme Q oxidoreductase), complex III (coenzyme Q-cytochrome c oxidoreductase), and complex IV (cytochrome c oxidase), coupled to electron transfer between the complexes. This proton motive force (PMF) is used to drive ATP production by complex V (F_1_F_0_-ATPase) as it transfers the accumulated protons in the intermembrane space back across the inner membrane [221]. Complex I transfers electrons from NADH, while complex II (succinate-coenzyme Q reductase or succinate dehydrogenase) transfers electrons from reduced flavin adenine dinucleotide (FADH_2_). Complex I and II transfer electrons to Coenzyme Q (also called ubiquinone in its oxidized form and ubiquinol in its reduced form), which passes the electrons to complex III. They are then carried to complex IV by Cyt c, with molecular oxygen being the ultimate electron acceptor at complex IV to form water. Various stressors can cause mitochondrial dysfunction, mitochondrial membrane depolarization, mitochondrial fission, and decreased energy production via oxidative phosphorylation. In this case, energy production may be produced via glycolysis instead, which is much less efficient and produces only 2 ATP rather than approximately 36 ATP per molecule of glucose [248,249]. Cancer cells display the Warburg effect, with an energy metabolism that is mainly through aerobic glycolysis [250,251,252].

The shift from using aerobic oxidative phosphorylation in the mitochondria to anaerobic glycolysis in the cytosol for energy production represents an adaptive survival mechanism to mitochondrial dysfunction or to aid in cell proliferation [250]. This was observed in vitro in several cancer (HepG2, HeLa, renal carcinoma A498, and prostate carcinoma PC3) cell lines as well as the non-cancer human embryonic kidney 293T (HEK293T) cell line using sublethal exposure levels of Ag NPs that did not significantly decrease the cell viability [253]. Lactate and pyruvate increased, indicating an increase in glycolysis; while ATP production decreased, mitochondrial membrane potential decreased, the activity of succinate dehydrogenase decreased, and lipid metabolism was inhibited, all of which indicate mitochondrial dysfunction. Additionally, the gene expression of oxidative phosphorylation related genes decreased, while the gene expression of glycolysis related genes increased [253].

#### 5.2.1. Effects of Nanoparticles on Energy Production In Vitro

A decrease in the activity of the ETC complexes, mitochondrial membrane potential, and an increase in indicators of mitochondrial dysfunction and mitochondrial mediated apoptosis have been observed in multiple in vitro studies. Decreased succinate dehydrogenase activity has been observed in human lymphocyte cells from peripheral blood treated with TiO_2_ NPs measured by the 3-(4,5-dimethylthiazolyl-2)-2,5-diphenyltetraxolium bromide (MTT) and 2-(4-Iodophenyl)-3-(4-nitrophenyl)-5-(2,4-disulfophenyl)-2H-tetrazolium (WST-1) cell viability assays, which depend on this mitochondrial complex II enzyme to metabolize the substrates to formazan dyes [254].

Treatment of mitochondria (isolated from a tumor produced with F10 melanoma cells transplanted in a C57 mouse) with single walled carbon nanotubes, multiwalled carbon nanotubes, and Fe_2_O_3_ NPs resulted in mitochondrial swelling, increased ROS, decreased mitochondrial membrane potential, decreased succinate dehydrogenase activity, activation of caspase-3, and Cyt c release [255]. Interestingly, these signs of mitochondrial disfunction and apoptosis were not observed in NP treated mitochondria isolated from healthy mouse skin tissue, indicating preferential toxicity to cancer cells by the NPs [256].

The enzyme activities of complex IV and complex V in the ETC decreased in A549 cells treated with a low exposure level of ZnO NPs, indicating mitochondrial dysfunction [257]. The cellular amount of anti-apoptotic protein Bcl-2 and the activity of the antioxidant enzyme SOD also decreased. Interestingly, these results were only observed in classical 2D monolayer cell culture and no changes were seen with the ZnO NP treatment in three-dimensional (3D) cell culture, except for a small increase in complex IV activity. The reason for this difference in results between the 2D and 3D cell culture may be due to the cells being in a more natural simulated environment in the 3D cell culture, with more extracellular matrix proteins, cell to cell adhesion, and intercellular signaling, giving them more natural function and increased resilience. Alternatively, it is possible that the increased extracellular matrix decreased the amount of ZnO NPs that were able to interact with the cells [257]. More specifically for complex IV, the protein expression of cytochrome c oxidase subunit II has been found to decrease in human U87 astrocytoma cells treated with cytotoxic levels of SiO_2_ NPs [258].

Treatment of mitochondria (isolated from male Wistar rat brain tissue) with Al NPs increased ROS production, depleted GSH, increased lipid peroxidation, induced mitochondrial swelling, and induced the release of Cyt c, indicating mitochondrial mediated apoptosis [259]. Since complexes I and III are the main sources of ROS formed as a by-product in the ETC, Arab-Nozari et al. (2019) [259] used substrates and inhibitors for ETC complexes I and III to determine that Al NPs were producing most of the ROS through inhibition of complex III [221].

Decreased levels of intracellular ATP were found in human oral squamous cell carcinoma (HSC-3) cells treated with Au NPs, while mitochondrial depolarization was seen with Ag NP treatment [260]. Silver NPs also lowered the mitochondrial membrane potential in pancreatic carcinoma (PANC-1) cells [261]. Additionally, mitochondrial swelling, decreased mitochondrial membrane potential, and uncoupling of oxidative phosphorylation were observed in mitochondria isolated from rat liver treated with Ag NPs [262]. The TiO_2_ food additive E171 induced a loss in mitochondrial membrane potential in human lung fibroblast (WI-38) cells [43].

#### 5.2.2. Effects of Nanoparticles on Energy Production In Vivo

The effect of Ag NPs and Ti NPs on mitochondrial function in vivo has been examined in male Wistar rats at a sub-chronic exposure level of 100 µg/kg/day by gavage for 21 days [255]. Treatment with both the NPs at once (total of 100 µg/kg/day) had more effect and was more toxic than treatment with either of the NPs individually at 100 µg/kg/day and resulted in swollen mitochondria in the rat livers with oxidative phosphorylation uncoupling. Additionally, the level of ROS increased, and the endogenous GSH antioxidant system was depleted and oxidized as seen with a decreased GSH/GSSG ratio [255].

Iron oxide NPs accumulated in the liver, heart, and circulatory systems of male albino mice, resulting in cardiac dysfunction [263]. The mice were treated with Fe_2_O_3_ NPs intraperitoneally once a week for 30 days at a dose of 25 mg/kg body-weight Fe_2_O_3_ NPs. Mitochondria isolated from the cardiac tissue were oxidatively damaged, with outer membrane damage, decreased ATP levels, increased ROS, increased ^●^NO, increased lipid peroxidation, increased protein carbonyl content, decreased antioxidant levels (SOD, CAT, GSH, GPx, GST, and vitamin C), and displayed mitochondrial mediated apoptosis through increased Bax, Cyt c, and cleaved caspase-3 protein expression [263]. Apoptosis is an ATP dependent process, and the level of cellular ATP is important in directing the cell towards either death by apoptosis or by necrosis [227]. Higher 50 mg/kg body-weight Fe_2_O_3_ NP treatment resulted in further depletion of ATP, less apoptosis, and an increase in necrosis [263].

### 5.3. Summary

Thus, a wide variety of NPs have been observed to produce mitochondrial dysfunction, trigger mitochondrial fission, impair various aspects of oxidative phosphorylation, and induce mitochondrial mediated apoptosis, with exceptions being NPs with intrinsic antioxidant properties. Nanoparticles affect different types of cells differently and to varying degrees. Depending on the situation, NP induced mitochondrial dysfunction and cell death may be detrimental, such as if this occurs in healthy cells and organisms; or may be beneficial and the toxicity used to advantage such as with the anticancer activity of various metallic NPs and NPs with antioxidant properties [264,265]. For example, in vitro studies comparing the toxicity of various NPs on cancer and noncancer cell lines have found Ag NPs [261], Au NPs [260], vanadium pentoxide NPs [266], lipophilic bismuth NPs [265,267], and graphene oxide NPs [37] to be preferentially toxic to cancer cells than to noncancer cells. A summary of the effects of NPs on mitochondrial function and energy production is shown in Table 5.

## 6. Effects of Nanoparticles on Insulin, Calcium, and Nitric Oxide Signaling Pathways

Nanoparticles have the potential to affect the functioning and regulation of biological systems either positively or negatively in many ways, including effects on insulin and blood glucose regulation, calcium signaling, and ^●^NO signaling.

### 6.1. Effects of Nanoparticles on Insulin Signaling

Insulin is an important anabolic peptide hormone that regulates the levels of key nutrients in the blood such as glucose, lipids, and amino acids, as well as having broader effects on growth, cognition, reproduction, gene expression, and longevity. Insulin is synthesized in the pancreas by beta cells, released into the blood stream in response to high nutrient levels in the blood, and binds to insulin receptors on cells inducing them to take in nutrients. The three insulin receptors (insulin receptor, type 1 insulin-like growth factor receptor (IGF-1R), and the orphan insulin receptor-related receptor (IRR)) are all part of the insulin receptor subfamily in the receptor tyrosine kinase superfamily [269]. Insulin-like growth factor 1 (IGF-1) protein is made in the liver and is involved in regulating cell proliferation, cell distribution, and tissue growth including bone growth and resorption [270]. Treatment of young and growing three-week old male Sprague-Dawley rats with 68, 203, and 610 mg/kg body-weight/day ZnO NPs for 28 days decreased the serum level of IGF-1 in the two higher treatment groups, reduced their body weight gain, detrimentally affected their bone growth, and decreased their bone density. The serum calcium to potassium ratio, normally balanced, decreased with the highest ZnO NP treatment indicating disrupted bone metabolism at this treatment level. Serum levels of aspartate aminotransferase (AST), alanine aminotransferase (ALT), and alkaline phosphatase (ALP) increased in the highest treatment group, indicating liver dysfunction, with the elevated ALP level also indicating bone damage. Zinc levels in the liver and bone tissue increased, potentially causing damage, and liver dysfunction might explain the lowered IGF-1 levels [270]. Osteoprotegerin (OPG) and receptor activator of nuclear factor-κB ligand (RANKL) are synthesized by osteoblasts and are involved in regulating bone resorption [271]. The ZnO NP treatment decreased the expression of OPG and decreased the OPG/RANKL ratio, indicating disruption of the OPG/RANK/RANKL/IGF-1 pathway and increased osteoclastic bone resorption [270].

Insulin-like growth factor binding protein-3 (IGFBP-3) is produced by a tumor suppressor gene and inhibits the binding of IGF-1 to IGF-1R, stopping them from encouraging cell proliferation and survival [37]. Graphene oxide NPs have been found to have anticancer effects and to be specifically toxic to the human bone osteosarcoma cell lines U2OS and SAOS2 compared to the non-cancer human osteoblast cell line hFOB1.19. Disruption of the insulin signaling pathway in U2OS cells by Clustered Regularly Interspaced Short Palindromic Repeats (CRISPR) and CRISPR-associated protein 9 (Cas9) knock-out of IGF-1 and IGFBP3 further exacerbated the cell death induced by the graphene oxide NPs, highlighting a potential weakness that could be targeted to increase the toxicity of graphene oxide NPs on osteosarcoma cells [37]. Reproductive toxicity is a concern with Ni NPs, and intrinsic apoptosis in rat testicular Sertoli-germ cells was induced by Ni NPs through upregulation of the apoptotic long-chain non-coding RNA LOC102551356, which regulates the expression of several genes including the upregulation of the gene and protein expression of IGFBP-3, leading to intrinsic apoptosis [272].

In insulin signaling, binding of insulin to the insulin receptor or binding of IGF-1 to the IGF-1R, results in the activation of various signaling cascades. One of these signaling cascades involves the recruitment and binding of insulin receptor substrate (IRS) proteins to the activated receptor. Insulin receptor substrate is subsequently phosphorylated, forming binding sites, allowing for the binding and activation of phosphatidylinositol 3-kinase (PI3K) among other effectors [273]. Activated PI3K leads to the phosphorylation and activation of Akt (also called protein kinase B (PKB)), which phosphorylates glycogen synthase kinase-3β (GSK-3β) and inactivates it. This inactivation stops GSK-3β from inhibiting glycogen synthase, thus allowing glucose to be stored as glycogen [274,275]. Activated Akt has many metabolic functions, including mediating the translocation of glucose transporter 4 (GLUT4) to the cell surface through the phosphorylation of 160 kDa Akt substrate (AS160), thus facilitating glucose uptake into the cell [276]. Another glucose transporter, glucose transporter 2 (GLUT2), remains on the cell membrane [277]. Akt and IRS-1 are regulated by 5′ adenosine monophosphate-activated protein kinase (AMPK), and phosphorylated AMPK activates insulin signaling [155]. Type 1 diabetes mellitus is primarily an autoimmune condition, where pancreatic beta cells are destroyed by the immune system and can no longer synthesize insulin, leading to hyperglycemia. Insulin is given to treat type 1 diabetes and control blood glucose levels [278]. Type 2 diabetes mellitus is a metabolic disorder caused by a combination of lifestyle, environmental, and genetic factors, where the cells develop decreased sensitivity to insulin and do not take in the glucose from the blood stream. Insulin resistance by the cells first results in excessive proliferation of the pancreatic beta cells in an effort to compensate and produce more insulin. This results in elevated oxidative stress and inflammation in the pancreatic beta cells, leading to exhaustion, dysfunction, decompensation, and potentially death of the pancreatic beta cells; as well as insulin deficiency [154]. Simply treating type 2 diabetes with insulin may not be sufficient due to the insulin resistance of the cells, and this aspect of the condition needs to be addressed in the treatment. Other drugs used to treat type 2 diabetes to lower the blood sugar such as metformin, sulfonylureas, thiazolidinediones, dipeptidyl peptidase-4 (DPP-4) inhibitors, and sodium-dependent glucose transporter 2 (SGLT2) inhibitors, have undesirable side effects, do not correct the underlying issues, and need to be continually taken [158]. Metformin and thiazolidinediones inhibit complex I in the mitochondrial ETC, inhibiting energy production, and raising the cellular adenosine monophosphate (AMP) to ATP ratio. This triggers liver kinase B1 (LKB1) to activate AMPK, which, among other processes, promotes insulin signaling through activating Akt and IRS-1 [155,279,280,281]. To address the underlying issues behind diabetes, good nutrition, a carefully controlled diet, adequate exercise, maintaining a healthy weight, and rectifying detrimental lifestyle and environmental issues are key. Supplementation with various antioxidants including vitamin C and vitamin E, and metals such as magnesium, zinc, chromium, and vanadium, have also been found to be beneficial in treating type 2 diabetes [277,278,282,283].

Several intriguing studies have found various antioxidant and metallic NPs that show promise in treating type 2 diabetes. Fullerene and gadofullerene NPs have superb antioxidant and anti-inflammatory activities. Gadofullerene NPs functionalized with β-alanine activated the IRS2/PI3K/Akt signaling pathway in db/db diabetic mice at 6 µmol/kg body-weight/day, decreased their hyperglycemia to normal blood sugar levels; and reduced the oxidative stress, inflammation, and dysfunction in their pancreatic islet cells. Encouragingly, these positive effects continued even after discontinuing the NP treatment, unlike the medications currently being used to treat type 2 diabetes [154].

Selenium is an essential micronutrient with antioxidant, anti-inflammatory, and detoxification properties. Oral treatment of high-fat-diet/streptozotocin-induced type 2 diabetic male Wistar rats with 0.4 mg/kg body-weight of Se NPs, either alone or in combination with metformin, decreased their fasting blood glucose level, lowered the insulin level, and normalized the levels of active phosphorylated IRS-1, phosphorylated Akt, phosphorylated GSK-3β, and phosphorylated AMPK in the insulin signaling pathway [155]. The Se NPs also decreased the blood serum level of liver enzymes ALT, AST, ALP, and gamma-glutamyl transferase (GGT) that were released due to diabetes-induced liver damage; decreased the serum lipid profile, were anti-inflammatory by modulating the cytokine expression, and restored the cellular antioxidant levels [155].

The transcription factor Nrf2 mediates the cellular response to oxidative stress by binding to the antioxidant response element in the promoters of genes for various antioxidant proteins and enzymes such as heme oxygenase-1 (HO-1), SOD1, CAT, GPx1, and NADPH quinone dehydrogenase 1 (NQO1), and inducing their expression [284]. Nuclear factor erythroid 2-related factor 2 is regulated by binding to Kelch-like ECH-associated protein 1 (Keap1) as well as by microRNA-27a, which is redox-sensitive. Quercetin is a bioflavonoid and an antioxidant. Quercetin-conjugated superparamagnetic iron oxide NPs induced antioxidant effects in streptozotocin-induced type 1 diabetic male Wistar rats treated with 25 mg/kg body-weight/day for 35 days. The gene expression of microRNA-27a was decreased, allowing increased gene expression of Nrf2, SOD1, and CAT. Treatment with bare superparamagnetic iron oxide NPs also decreased the expression of microRNA-27a and increased the expression of Nrf2, but to a lesser extent [285]. In another study, treatment of high-fat-diet/streptozotocin-induced type 2 diabetic Sprague-Dawley male rats with superparamagnetic iron oxide NPs at 22, 44, and 66 µmol Fe/kg body-weight once a week for four weeks normalized their fasting blood glucose level, lowered their insulin level, and corrected their blood lipid profile [275]. The levels of phosphorylated insulin receptor, PI3K, and phosphorylated GSK-3β increased in the liver tissue, indicating increased insulin signaling. This induced increased levels of GLUT2 in the liver and pancreas and GLUT4 in the adipose tissue, indicating increased glucose uptake and decreased insulin resistance. Interestingly, the effects of treatment with superparamagnetic iron oxide NPs were comparable to treatment with metformin [275].

Due to their involvement in metabolic processes, zinc NPs are being explored for treating type 2 diabetes. Zinc is an essential trace mineral and is necessary for the function of over 300 enzymes, and disrupted zinc homeostasis is linked to diabetes. A large proportion of people suffering from type 2 diabetes are deficient in zinc, and zinc supplementation has been found to have positive effects [278]. Several studies have examined the effect of ZnO NPs in diabetic models. Treatment of streptozotocin-induced diabetic male albino rats with 10 mg/kg body-weight/day ZnO NPs for 30 days, or 10 mg/kg body-weight/day Ag NPs for 30 days, had protective effects and normalized the fasting blood sugar, insulin, and insulin resistance score levels [286]. Additionally, the ZnO NP treatment successfully ameliorated diabetic vascular complications [287]. In a similar study, Sprague-Dawley rats with streptozotocin-induced diabetes treated with 10 mg/kg body-weight/day ZnO NPs for 30 days resulted in more normalized blood glucose and insulin levels, as well as increasing the GLUT2 and glucokinase expression in the liver tissue.

Similar effects, but to a lesser degree, were observed with Ag NP treatment at 10 mg/kg body-weight/day for 30 days [277]. High dose oral treatment of healthy male Sprague Dawley rats with Ag NPs at 50 and 100 mg/kg body-weight/day for 90 days resulted in increased levels of IRS-1, Akt, GSK3β, and mammalian target of rapamycin (mTOR) in the liver tissue, increased the activity of SOD and CAT, and increased lipid peroxidation levels in response to oxidative stress [288].

Regarding TiO_2_ NPs, treatment of male CD-1 mice with anatase TiO_2_ NPs at 64 and 320 mg/kg body-weight/day for 14 weeks was detrimental, raised the blood glucose level without changing the blood insulin level, induced oxidative stress and inflammation, and showed disruption of the insulin signaling pathway with increased IRS-1 phosphorylation and decreased phosphorylation of Akt [289]. Additionally, ingestion of the TiO_2_ food additive, T171, at 5 mg/kg body-weight/day by BALB/c mice resulted in altered gene expression in the mouse colon cells for several insulin related genes [290]. Microarray analysis of human colon carcinoma Caco-2 cells treated with T171 altered gene expression in the insulin processing pathway [42].

Platinum NPs increased the activation of Akt in HepG2 cells at subtoxic treatment levels, potentially indicating perturbations in the IGF-1 signaling pathway [26].

Many people with type 2 diabetes also develop non-alcoholic fatty liver disease. In a study simulating insulin resistance and non-alcoholic fatty liver disease with high-fat-diet fed obese C57BL/6 mice, treatment of the mice with ZnO NPs at 0.5 mg/kg body-weight every second day for 15 days was found to be beneficial and to decrease insulin resistance and hepatic steatosis through a signaling pathway involving Silent mating type Information Regulation 2 homolog 1 (SIRT1) (which has two zinc finger motifs that require Zn^2+^ binding for activity), LKB1, and AMPK [279]. Activation of AMPK results in the phosphorylation of the transcription factor sterol regulatory element binding protein-1c (SREBP-1c), which keeps SREPB-1c from translocating to the nucleus and inducing lipogenic gene expression, thus serving to relieve the dysregulated hepatic lipogenesis that is characteristic of non-alcoholic fatty liver disease [279].

The most common type of neurodegenerative dementia is Alzheimer’s disease, which involves oxidative stress, neuroinflammation, and insulin resistance in the brain tissue [203]. In fact, this insulin resistance and impaired insulin signaling in the brain has been termed type 3 diabetes [291]. Gold NPs have anti-inflammatory and antioxidant effects in biological systems and showed promising results for the treatment of dementia in an experimental model for sporadic Alzheimer’s disease created by intracerebroventricular-streptozotocin injection in male Wistar rats. Treatment of the rats with Au NPs at 2.5 mg/kg body-weight prevented the induced neuroinflammation, oxidative stress, memory deficits, and impairment of ATP production seen in the sporadic Alzheimer’s disease rats [203]. In pancreatic RIN-5F cells that had been treated with the diabetogenic plasticizer di(2-ethylhexyl) phthalate (DEHP) to induce a diabetic state, Au NPs stabilized with the medicinal coumarin, wedelolactone, had anti-diabetic activity, improved insulin sensitivity, and improved glucose uptake. This was seen with increased insulin secretion, decreased apoptosis, and increased gene and protein expression of IRS-1, GLUT2, and the insulin receptor. Additionally, treatment with the wedelolactone stabilized Au NPs reduced lipid peroxidation and normalized the levels of the antioxidant enzymes SOD, CAT, and GPx [284].

### 6.2. Effects of Nanoparticles on Calcium Signaling

Calcium ions (Ca^2+^) are essential signaling molecules in cells and are involved in many processes such as muscle function, nervous system function, skeletal mineralization, blood coagulation, cell proliferation, apoptosis, and enzyme activity [292,293]. They are the most abundant cation and the fifth most abundant element in the human body [293,294]. The level of calcium in the blood is tightly regulated between 2.1–2.6 mM, with alteration by at most 3% [293]. Intracellular calcium homeostasis is also tightly regulated by the ER and mitochondria [247]. Endoplasmic reticulum stress causes calcium that is stored in the ER to be released into the cytosol. Stress can also cause an influx of extracellular calcium into the cytosol through plasma membrane voltage-gated calcium channels such as L-type, N-type, and T-type calcium channels [292]. Mitochondria take up calcium, increasing the mitochondrial calcium levels and potentially triggering mitochondrial dysfunction and mitochondrial mediated apoptosis [295,296]. Thus, the effects of NPs on intracellular calcium levels and signaling is of great interest.

Silicon dioxide NPs, TiO_2_ NPs, and carbon NPs all disrupted calcium homeostasis in rat pulmonary artery smooth muscle cells (PASMCs). Acute exposure with each of these NPs resulted in increased intracellular calcium, with the SiO_2_ NPs raising the level the most, and the TiO_2_ NPs raising the level the least [297]. The SiO_2_ NP induced influx of calcium was from both intracellular reserves in the sarcoplasmic reticulum and from extracellular sources brought in via voltage-gated calcium channels and transient receptor potential vanilloid (TRPV) channels in the plasma membrane. The sarcoplasmic reticulum is the main intracellular calcium reserve in PASMCs, and inhibitors were used to determine that ryanodine receptor channels in the sarcoplasmic reticulum and not inositol 1,4,5-triphosphate (IP_3_) receptor channels were involved in this release. Ryanodine and IP_3_ receptors in the sarcoplasmic reticulum and ER are part of the calcium induced calcium release (CICR) mechanism that amplifies small intracellular calcium increases from extracellular signaling that activated plasma membrane calcium channels. It is unclear how NPs activate plasma membrane calcium channels, with possibilities being through direct interaction or through altering the membrane potential. Deregulation of calcium signaling is found in hypertensive disease, and PASMCs from rats with hypoxic-induced pulmonary hypertension had a larger increase in intracellular calcium from SiO_2_ NP treatment than did PASMCs from healthy rats. Oxidative stress due to the SiO_2_ NP treatment is involved in the rise in intracellular calcium, and the antioxidant N-acetyl-cysteine was able to prevent this rise [297]. The involvement of IP_3_ receptors in the release of calcium from the ER has been reported in SH-SY5Y cells in response to Ag NPs [295]. The amount of close contact between the mitochondria and ER increased (termed mitochondria-associated ER membranes (MAMs)), with this facilitating calcium signaling and the uptake of calcium into the mitochondria from the ER. Additionally, phosphatase and tensin homolog deleted on chromosome ten (PTEN) localized from the cytoplasm to the ER and MAMs in Ag NP treated cells, where PTEN interacted with IP_3_ receptors, activating them via dephosphorylation, and thus increasing calcium release and intracellular calcium levels [295]. In a follow-on study with Ag NP treatment on SH-SY5Y cells and oral treatment of Sprague-Dawley rats for 28 days, Ag NPs induced ER stress and increased intracellular calcium levels, which activated calmodulin-dependent protein kinase kinase β (CaMKKβ), which in turn activated AMPK, which inhibited mTOR signaling, ultimately resulting in autophagy [298]. An increase of intracellular calcium has also been found with carbon black FW2 NPs [49], CuO NPs [299], and nickel oxide (NiO) NPs [300]. Treatment of zebrafish embryos with Ag NPs resulted in the down-regulation of calcium signaling [301].

Another mechanism for NP induced intracellular calcium increase is through inhibited enzyme activity of plasma membrane calcium and sodium ion exchangers. Calcium ATPases (Ca^2+^-ATPases) and calcium-magnesium ATPases (Ca^2+^/Mg^2+^-ATPases) are located on the plasma membrane and remove calcium from the cell to sustain the cellular calcium gradient [247]. Sodium-potassium ATPases (Na^+^/K^+^-ATPases) are integral plasma membrane proteins that remove three sodium ions for every two potassium ions that they take in. These enzymes are powered and activated by ATP and have inhibited function in situations of mitochondrial dysfunction and lowered ATP production. Additionally, increased ROS inhibits the enzyme activity of calcium and sodium ion exchangers leading to ion imbalances and a buildup of calcium in the cell. Treatment of HUVECs with Si NPs triggered mitochondrial ROS, increased the level of intracellular calcium, induced mitochondrial dysfunction, decreased ATP production, and decreased the enzyme activities of Ca^2+^-ATPase, Ca^2+^/Mg^2+^-ATPase, and Na^+^/K^+^-ATPase [247].

Nanoparticles have been found to activate N-methyl-D-aspartate (NMDA) receptors in the central nervous system. These ligand-gated cation channels are a class of ionotropic glutamate receptors (iGluRs) that, once activated by both glutamate and glycine (or D-serine instead of glycine), mediate excitatory neurotransmission at glutamatergic synapses and allow a large influx of calcium into the postsynaptic neuron. N-methyl-D-aspartate receptors have many subunits including NMDA receptor 1 (NR1), NR2A, NR2B, NR2C, NR2D, NR3A, and NR3B [302,303]. Silver NPs activated the NMDA receptors in primary rat cerebellar granule cells (CGCs) and increased the intracellular calcium level, which led to mitochondrial dysfunction, decreased mitochondrial membrane potential, and increased oxidative stress [296]. Interestingly, zinc inhibits the opening of NMDA receptors, and supplementation with zinc chloride (ZnCL_2_) protected the Ag NP treated neurons from overactivation of the NMDA receptors, excessive calcium influx, and intracellular imbalance [304]. Low non-toxic Ag NP treatment of human embryonic stem cell-derived glutamatergic neurons decreased the protein expression of NMDA receptor subunits, possibly as a protective response by the cell [305]. Furthermore, in in vivo experiments, acute intraperitoneal injection of male Wistar rats with magnesium oxide (MgO) NPs decreased the level of glutamate and increased the gene expression of NMDA subunit NR2B in the hippocampus [303]. Acute intraperitoneal injection of ZnO NPs also decreased the level of glutamate in the hippocampus. An increase in the gene expression of NMDA subunit NR2A was seen but was not statistically significant [303]. Similarly, oral treatment of male Wistar rats with Ag NPs increased the gene expression of NMDA receptor subunits NR1, NR2A, and NR2B in the brain tissue. Interestingly, co-treatment with the antioxidant bioflavonoid rutin (quercetin-3-O-rutinoside) had protective effects [306].

Depending on their composition, NPs can be designed to restore intracellular calcium balance [307]. N-acetylglucosamine is used in the post-translational modification of several calcium regulating proteins, and so was chosen to be coated onto biodegradable and biocompatible polyketal NPs for intracellular release during degradation of the NP. These NPs, as well as N-acetylglucosamine decorated polyketal NPs carrying the calcium modulating protein S100A1, were able to decrease arrhythmogenic calcium release and improve sarcomere function in cardiac myocytes isolated from male Sprague-Dawley rats with induced nonischemic heart failure [307].

### 6.3. Effects of Nanoparticles on Nitric Oxide Signaling

Nitric oxide is an extremely important gaseous free-radical signaling molecule that is produced by tissues such as endothelial and nerve cells and is also released from immune cells as a killer molecule [123]. In fact, the 1998 Nobel Prize in Physiology or Medicine was awarded to Ferid Murad, Robert Furchgott, and Louis Ignarro for their discoveries regarding this signaling molecule [123]. Nitric oxide has no electrical charge and is soluble in both hydrophobic and hydrophilic biological environments, enabling it to passively diffuse across biological membranes [308]. Due to its unpaired electron, ^●^NO reacts with transition metals with unpaired electrons (such as ferrous iron (Fe^2+^) and Cu^+^), and with free radicals (such as superoxide forming peroxynitrite (ONOO^−^)). Nitric oxide has a short biological lifespan and generally reacts and is used within 2 s [308]. It is involved in regulating many biological processes such as neuronal signaling, vasorelaxation, bronchodilation, vascular permeability, angiogenesis, platelet aggregation, inflammation, wound healing, hormone production, gastrointestinal tract motility, gene expression, and inflammation [309,310]. Decreased levels of ^●^NO are associated with various disease states such as cardiovascular disease, respiratory disorders, impaired immune response, and cancer [309]. Prolonged elevation of the ^●^NO level becomes toxic and is also associated with diseases such as cancer and inflammatory conditions [123,311]. The level of ^●^NO is controlled by the three isoforms of nitric oxide synthase (NOS) enzymes, which oxidize L-arginine to N^G^hydroxy L-arginine, which is then oxidized to L-citrulline, producing ^●^NO. Neuronal NOS (nNOS or NOSI) is constitutively expressed in the nervous system and is also inducible in some situations. Inducible NOS (iNOS or NOSII) expression is stimulated by the immune system when needed but has also been found to be constitutively expressed in various tissues. Endothelial NOS (eNOS or NOSIII) was discovered third and is constitutively expressed in the endothelium and is also inducible in some situations [123]. The NOS enzymes require the simultaneously binding of calcium bound calmodulin, heme, tetrahydrobiopterin, FAD, and flavin mononucleotide (FMN) for their activity. Stimulated iNOS produces a sustained long-term production of ^●^NO, while nNOS and eNOS transiently respond to stimulation by only producing ^●^NO for a short time. In the nervous system, activation of the NMDA receptors raises the intracellular calcium levels, allowing calcium to bind to calmodulin and activate nNOS. Elevated intracellular calcium levels also trigger eNOS to produce ^●^NO, while iNOS is regulated through its gene expression rather than by intracellular calcium levels [123].

Many different NPs have been found to effect ^●^NO signaling. Contraction of the airway smooth muscles is regulated through the G protein-coupled muscarinic receptor by various signaling molecules, including ^●^NO and acetylcholine [311]. Silver NP treatment of rat smooth muscle tracheal rings that had been pre-treated with acetylcholine increased iNOS protein expression and ^●^NO production, activated the muscarinic receptors, and induced contraction [310]. In a follow-on study using only one tracheal cell type, airway smooth muscle (ASM) cells isolated from adult male Wistar rats, Ag NP treatment increased iNOS expression, increased ^●^NO production, and activated the muscarinic receptors independently of acetylcholine stimulation [311]. Increased iNOS expression and ^●^NO production with Ag NP treatment have also been found in SVEC4-10 mouse endothelial cells [312] and in human fetal osteoblast cells (hFOB 1.19) [313]. Additionally, Ag NPs increased calcium levels; and the protein expression of calmodulin, iNOS, and nNOS in human embryonic stem cell-derived glutamatergic neurons [305]. Increased gene expression of iNOS and eNOS were found in Ag NP treated chicken Sertoli cells [314]. Interestingly, Ag NPs have been found to have beneficial anti-inflammatory effects and decreased ^●^NO and iNOS in LPS-induced RAW 264.7 cells [168]. However, in a hypertension model, high dose Ag NP treatment of isolated perfused hearts from spontaneously hypertensive rats resulted in vasoconstriction and increased contraction of the cardiac tissue. In this experiment, Ag NP treatment resulted in increased ROS, decreased ^●^NO production, decreased iNOS and eNOS expression, but increased expression of phosphorylated eNOS [315]. Increased phosphorylation of eNOS has generally been used as an indication of eNOS activation; however, this is not always the case, and in some circumstances increased eNOS phosphorylation does not result in an increase in ^●^NO production [316]. Decreased iNOS gene expression was also observed with Ag NP treatment of C57BL/6J male mice with high-fat Western diet induced metabolic syndrome [317].

Gold NPs have been found to have beneficial anti-inflammatory effects and to decrease pro-inflammatory mediators such as ^●^NO and iNOS in LPS-induced RAW 264.7 cells [150,168,218,318] and in mouse BV-2 microglial cells [166]. Gold NPs also decreased vascular endothelial growth factor (VEGF)-induced eNOS phosphorylation in rhesus macaque choroid-retinal endothelial RF/6A cells [312].

Generally, TiO_2_ NPs have been found to have detrimental, inflammatory, oxidative, and apoptotic effects, and to affect the ^●^NO production both in vitro [173,194,319] and in vivo [320,321,322,323]. However, Gholinejad et al. (2019) [324] observed no change in iNOS and eNOS expression in HUVECs treated with TiO_2_ NPs. Additionally, TiO_2_ nanotubes (4–10 nm in diameter and 100–500 nm in length) have been found to decrease the protein expression of eNOS via down-regulation of the eNOS transcription factors kruppel-like factor 2 (KLF2) and KLF4, and to decrease the ^●^NO level in HUVECs [325].

Zinc oxide NPs have been found to have positive effects and to help normalize iNOS expression and ^●^NO production in several disease models such as inflammation with LPS-induced RAW 264.7 cells [326], diabetes with streptozotocin-induced diabetic male albino rats [287], and bacterial infection with nontypeable *Haemophilus influenzae* infected RAW 264.7 cells and C57BL/6 mice [327]. Other in vitro studies not simulating disease states have found ZnO NP treatment to induce toxicity and increase iNOS and ^●^NO levels [137,328,329]. Zinc ferrite (ZnFe_2_O_4_) NPs have been found to have a similar effect and increase iNOS expression [330].

Copper NPs have been found to have beneficial effects at low dose exposure levels, but toxic effects at high dose exposure levels. Low dose Cu NP treatment of male and female Wistar albino rats with induced myocardial infarction at 1 mg/kg body-weight/day for 4 weeks was beneficial and decreased the induced myocardial damage, ROS, and inflammation; as well as normalizing the serum ^●^NO level. Interestingly, addition of swimming exercise further enhanced this protective effect by low dose Cu NP treatment [331]. Conversely, high dose Cu NP treatment of male Sprague-Dawley rats at 400 mg/kd body-weight/day for 7 days had toxic effects and increased the levels of ROS, iNOS, and ^●^NO in the liver tissue [332]. Treatment of LPS-induced primary macrophages from C57BL/6 mice with Cu NPs normalized the LPS-induced ^●^NO level. Interestingly, treatment with arginase inhibitors as well as Cu NPs inhibited the Cu NPs from decreasing the LPS-induced ^●^NO level. This indicates that Cu NPs may decrease the level of induced ^●^NO by activating arginase, and thus limiting the availability of L-arginine for ^●^NO synthesis [333]. A decrease in LPS-induced ^●^NO level with Cu NP treatment has also been seen in primary mouse macrophages from bone marrow [74].

Cerium dioxide NPs decreased both LPS-induced sepsis and cecal inoculum induced sepsis iNOS gene and protein expression in male Sprague Dawley rats treated intravenously with 0.5 mg/kg body-weight CeO_2_ NPs [36,213]. Additionally, CeO_2_ NP treatment decreased LPS-induced inflammatory cytokines, ROS, iNOS protein expression, and ^●^NO production in Kupffer cell macrophages extracted from rat liver from male Sprague Dawley rats with LPS-induced sepsis [213]. Mesenteric arterioles isolated from healthy male Sprague Dawley rats injected with 0.1 mg CeO_2_ NPs also exhibited decreased ^●^NO production; however, in this experimental model ROS increased and microvascular function was disrupted [35].

Platinum NPs have been found to decrease the levels of inflammatory mediators in LPS-induced RAW 264.7 cells, including decreasing ^●^NO and iNOS expression [334,335]. However, treatment of human acute myeloid leukemia THP-1 macrophages with anisotropic lycopene capped Pt NPs had toxic effects and increased oxidative stress and ^●^NO levels [25].

Superparamagnetic iron oxide NPs at a treatment level of 80 µg/mL for 24 h reduced LPS-induced inflammatory mediators and decreased iNOS expression in RAW 264.7 cells [117]. However, in non-LPS-induced RAW 264.7 cells, superparamagnetic iron oxide NPs increased the ^●^NO level and oxidative stress, indicating increased toxicity [336,337].

Silica NPs had toxic effects and increased inflammation, oxidative stress, and ^●^NO levels in the liver and kidneys of male Wistar rats treated with 25, 50, 100, and 200 mg/kg body-weight/day for 30 days [338]. Similarly, treatment of HUVECs with Si NPs resulted in inflammation, oxidative stress, and dysfunction in the endothelial cells, with increased ^●^NO levels, increased iNOS expression, and decreased eNOS expression [339]. Endothelial dysfunction was also observed in another study with Si NP treatment of HUVECs; however, the ^●^NO level decreased, the expression of total NOS and eNOS decreased, and the expression of iNOS increased [340].

Carbon NPs come in various forms and have varying effects in biological systems. Nanodiamond had a neuroprotective effect in an aluminum-induced Alzheimer’s disease model in male albino Wistar rats and decreased the aluminum-induced iNOS expression [334]. Decreased ^●^NO production was found in HepG2 cells treated with graphene quantum dots [335], while graphene oxide nanosheets did not significantly change the ^●^NO production in neuroblastoma NB41A3 cells [341]. Multi-walled carbon nanotubes (20–30 nm outer diameter and 10–30 µm in length) increased iNOS expression in RAW 264.7 cells [342].

Various other NPs have also been found to affect ^●^NO signaling. Selenium NPs capped with *Ganoderma lucidum* polysaccharides had anti-inflammatory effects and decreased the LPS-induced ^●^NO level in RAW 264.7 cells [156]. Manganese dioxide NPs had antioxidant and chondroprotective effects and decreased the ^●^NO level in cytokine-challenged cartilage explants [343]. Chitosan NPs exhibited an antioxidant effect and normalized the serum ^●^NO level in male albino Wistar rats with diethyl nitrosamine induced hepatocellular carcinoma [344]. Hydroxyapatite NPs decreased ^●^NO production and the protein expression of phosphorylated eNOS in HUVECs [345]. Finally, propylene glycol alginate sodium sulfite NPs had antioxidant effects, inhibited myocardial damage in streptozotocin-induced diabetic male Wistar rats, and inhibited the diabetes-induced decrease in NOS activity and ^●^NO level in the myocardial tissue [346].

Table 6 shows a summary of the effects of various nanoparticles on insulin, calcium, and nitric oxide signaling pathways.

## 7. Nanoparticle–Protein Interactions

### 7.1. Interactions of Nanoparticles with Zinc-Dependent Proteins

Zinc is a trace metal that plays crucial roles for cell growth, cell division, development, DNA synthesis, RNA transcription, differentiation, and in the maintenance of protein structure and stability. Additionally, zinc has a vital role in the regulation of signaling pathways in both the innate and adaptive immune system and is an antioxidant [351,352,353,354,355]. Many proteins have zinc-finger motifs (including proteins in approximately 10% of the human genome) that may be bound by zinc [353]. Since zinc is not stored in the human body, it must be obtained through diet from foods such as oysters, beef, chicken, turkey, egg yolks, and nuts [352,356].

Many of the immune cells depend on proper zinc homeostasis for normal functioning [355], and disruption in zinc homeostasis can result in inflammation, decrease of and impairment of T and B lymphocytes and natural killer cells, reduction in natural killer cell activity, and increase in cytotoxicity to monocytes [357,358]. Interestingly, the lack of taste and smell sometimes associated with coronavirus disease 2019 (COVID-19) infection is an indication of a severe zinc deficiency resulting from the high need for zinc by the immune system in fighting the infection [359].

Exposure to NPs may disturb zinc homeostasis, as was found with Ag NPs which increased intracellular zinc levels in CGCs [304,360], and CuO NPs which inhibited the zinc metalloenzyme histone deacetylase (HDAC) as well as the activity of different classes of HDAC proteins in A549 cells [361].

The two major families of zinc transporters are Zinc Transporter (ZnT) and Zinc-regulated, Iron-regulated transporter-like Protein (ZIP) [351]. Zinc oxide NPs resulted in increased gene expression of zinc transporters genes ZnT1 and ZnT2 in HepG2 cells [362]. Increased protein expression of ZnT1 and ZIPs as well as an increase of cytosolic zinc levels was seen in SH-SY5Y cells treated with ZnO NPs [363]. One study using molecular docking found that ZnO NPs interact with glycine residues on proteins via hydrogen bonding [364].

### 7.2. Interactions of Nanoparticles with Copper-Dependent Proteins

Copper is an essential trace element that is crucial for proper organ function and a variety of metabolic processes [365]. Copper is important for various processes in the nervous system such as synaptic transmission, axonal targeting, growth of axons or dendrites, and modification of signaling cascades [366]. It acts as a cofactor for a variety of enzymes such as cytochrome c oxidase (CcO), tyrosinase, and nitrite reductase. Additionally, copper participates in signaling pathways, protein secretion, and cell division [367,368]. Copper is acquired from the diet and is transported and stored in the body by ceruloplasmin and albumin [368]. Foods that are high in copper include: legumes, mushrooms, chocolate, nuts, and seeds [369].

Excess copper can inhibit various proteins such as acetylcholinesterase (AChE), succinate dehydrogenase, and various proteases, esterases, lipases, and glucosidases. The effect of Ag NPs on AChE was observed using CD and was found to disrupt its secondary structure and decrease AChE activity [368]. Copper can inactivate enzymes by binding to cysteine residues [366]. Copper poisoning is rare, however, when it does occur it results in liver and kidney damage which may lead to organ failure [368]. On the other hand, low levels of copper in the body are detrimental and may disrupt the immune system. Copper oxide NPs were found to regulate copper chaperone proteins in renal cell carcinoma cells and disrupted copper transportation [301].

Silver ions may competitively bind to various proteins in the place of copper ions [370]. It has been found that Ag ions bind to cysteine residues on proteins that contain Cu-binding sites, resulting in potential disruption of protein function [371]. Silver NPs have been found to competitively bind to copper transporters, resulting in a copper deficiency in *Drosophila melanogaster* [372]. Copper homeostasis was found to be disrupted in rats exposed to silver chloride through diet [367]. In a study where mice were treated with concentrations of 10 μg/g body-weight Ag NPs over four days, it was thought that Ag ions released from the Ag NPs disrupted their copper balance [371].

### 7.3. Interactions of Nanoparticles with Iron-Dependent Proteins

Iron is an essential trace element that is crucial for the function of various proteins involved in oxygen transport and sensing, metabolism, detoxification, and iron storage and transportation [373]. Since iron levels in macrophages affects the production of cytokines, iron plays an important role in the immune response [374]. Iron in the human body is either bound to heme or is present as ionic iron [375]. Hemoglobin in red blood cells binds most of the iron in the body, with approximately a quarter of the total iron stored in hepatocytes and macrophages in the spleen and kidney [374]. Most of the body’s iron requirement is met by reusing iron from old erythrocytes [374,376]. Iron deficiency is a common condition that often remains undiagnosed [377].

The composition of the NP can have different biological effects. Silver NPs have been found to interact with cellular iron and produce ROS in HepG2 cells [378], while silver ions are able to disrupt iron homeostasis [171]. Spinel zinc ferrite (ZnFe_2_O_4_) NPs affected iron homeostasis signaling in rat alveolar macrophages (NR8383); however, ZnO NPs tested at the same concentrations had no significant effect [379].

Ferroptosis occurs due to a large amount of iron accumulation resulting in cell death [380]. Nanoparticles have been shown to induce ferroptosis in cells [381]. In particular, ZnO NPs have been seen to induce ferroptosis in HUVECs, increase ROS, disrupt iron homeostasis by affecting uptake, storage, and export of iron, and upregulate iron transporter protein genes resulting in accumulation of iron in the cells [373].

Iron-binding proteins include Hb, myoglobin, cytochromes, NADPH, succinate dehydrogenase, frataxin, transferrin, ferritin, GST, and ferrochelatase [374,375,382]. Transferrin adsorbs to Au NPs, resulting in changes to the secondary structure [374,375,383,384]. Magnetic NPs prepared using Fe (III) and Fe (II) were found to absorb onto lactoferrin [385].

Cytochrome c is a heme protein in mitochondria that plays an essential role in the ETC and apoptosis [386,387]. Several studies have found upregulation of Cyt c in various cell lines and animal models, potentially due to apoptosis caused by NPs or by a different response [52,230,240,256,259,263,347]. A molecular dynamics (MD) simulation found that the adsorption of Cyt c to Si NPs was through hydrophobic interactions between key amino acid residues and the Si NPs resulting in structural stabilization. The active site of Cyt c became inaccessible to ligands after adsorption of Si NPs [387]. Using UV-vis, fluorescence, and resonance Raman (RR) spectroscopy, it was found that Si NPs absorb onto Cyt c through electrostatic interactions resulting in conformational changes and modifications to the active site of Cyt c [388].

Iron oxide NPs absorb electrostatically to Cyt c, however, no absorption occurs with PEG-coated iron oxide NPs. After adsorption to iron oxide NPs, the heme group of Cyt-c is reduced while adsorption to PEG-coated iron oxide NPs did not result in reduction of the heme group [389]. Using DLS, zeta potential measurements, static and synchronous fluorescence spectroscopy, CD spectroscopy, and ultraviolet–visible (UV–vis) spectroscopy, it was found that Fe NPs interact with Cyt c through hydrogen bonding and van der Waals forces. It was also found that conformational changes to the tertiary structure, changes in heme position, and unfolding in Cyt c were observed. Through absorption spectroscopy, it was observed that the Fe NPs disrupt the Fe…S (methionine 80) bond [390].

Manganese NPs induced conformational changes to the tertiary structure of Cyt c near phenylalanine residues via hydrophobic interactions [391].

Hemoglobin is an important protein for oxygen transport in the body. Hemoglobin is a tetrameric protein where each subunit has a heme group consisting of a porphyrin ring with a Fe ion for binding oxygen [392]. The interaction between NPs and Hb is important to study as the interactions may disrupt transport of oxygen by red blood cells in the body [254]. It is unclear whether NPs induce structural changes in Hb [393]. Adsorption of porcine Hb in its oxygenated form to Si NPs was studied using UV- vis spectrophotometry, CD spectroscopy, and oxygen binding measurement. It was found that there was a significant loss of the secondary structure of the Hb protein, the heme group was intact, and enhanced oxygen affinity after adsorption to Si NPs [20]. A follow-on study found similar observations using human Hb, and it was determined that the increased oxygen affinity was due to changes in the structure of the Hb proteins and interaction with the Si NPs [394]. Adsorption of bovine Hb to Si NPs induced structural changes and heme degradation, where hydrophobic Si NPs were found to induce a greater change than hydrophilic Si NPs [395]. Hydrophobic interactions were found to be involved in the adsorption of human Hb to SiO_2_ NPs, and temperature affected the binding affinity of the NPs to the Hb [360]. Furthermore, SiO_2_ NPs induced structural changes, heme degradation, and release of iron from Hb [124].

Hydrogen bonding and van der Waals forces were found to be involved in the interaction between TiO_2_ NPs and human Hb. Titanium dioxide NPs bound to Hb in a dose-dependent manner in human erythrocytes treated with concentrations of 300 mg/mL or higher [254]. It has also been found that TiO_2_ NPs induced structural changes in human Hb as well as cleavage of the heme group. This study used DLS, TEM, and CD and concentrations varying from 0–200 µg/mL TiO_2_ NPs. It is thought that the degradation of the heme group is due to ROS formation induced by exposure to TiO_2_ NPs [396]. Using FTIR spectroscopy, fluorescence spectroscopy, and molecular docking, Hb structure was determined to not significantly change after exposure to concentrations of 0–10 μM TiO_2_ NPs [397].

Zero valent iron NPs have been found to induce structural changes in the tertiary structure of human Hb via CD and UV-vis spectroscopy. Displacement and degradation of the heme was also observed with concentrations of 0–40 μM of ZVFe NPs [80]. Zinc oxide NPs induced conformational changes in Hb and were seen to bind to tryptophan and the heme-porphyrin groups on human Hb via hydrophobic and electrostatic interactions [398].

Silver NPs have been found to slightly alter the secondary structure of human Hb [399]. Using CD and UV-vis spectroscopy, the interaction between Ag NPs and human Hb was found to be concentration- and time-dependent. Concentrations of 0–0.001 M Ag NPs bound to the heme and induced structural changes in Hb [400]. Silver NPs were found to bind to bovine Hb and induce both secondary and tertiary structural changes [401]. Using Fourier-transform infrared (FT-IR) spectroscopy, adsorption of Hb to both Ag NPs and Au NPs occurred, however, no changes to the secondary structure of Hb were observed for either of the NPs [402]. Spherical Au NPs were found to bind to human Hb stronger than star-shaped Au NPs, and binding to Hb occurred via hydrogen bonding and van der Waals forces, with no changes in the secondary structure [403].

Cerium dioxide NPs were found to induce degradation of heme as well as structural changes to human Hb using concentrations of 0–27.5 μM CeO_2_ NPs [404]. Dose-dependent changes were induced in the secondary structure of human Hb after interacting with CeO_2_ NPs via hydrogen bonding, carbon hydrogen bonding, and electrostatic interactions [405]. However, it has also been found that no significant changes to the structure of human Hb occur using concentrations of 2.5–27.5 μM [406].

Graphene oxide nano-sheets at concentrations of 0–10 μg/mL were found to interact with human Hb and denature the quaternary and secondary structure [407]. Copper NPs and Cu-Zn alloy NPs induced aggregation of human Hb [408]. Aluminum oxide NPs at concentrations of 0.02–2 mg/mL induced quaternary structural changes in human Hb [409]. Nickel oxide NPs induced significant changes to the secondary and quaternary structure of Hb, and displaced various aromatic residues and the heme group [410].

### 7.4. Interactions of Nanoparticles with Calcium-Dependent Proteins

Calcium is an essential nutrient that is vital to normal physiological functioning by contributing to the maintenance and formation of the skeletal system, regulation of hormones, muscular and vascular contraction and dilation, transmission of nerve impulses, and enzyme activity [292,411,412]. Calcium also aids in regulating blood pressure and cholesterol levels [413]. Intracellular calcium ions are mainly stored in the ER, and release of calcium from the ER is integral to calcium signaling [292]. Dairy products, nuts and seeds, and dark green leafy vegetables are excellent sources of calcium [413]. Parathyroid hormone and calcitonin help maintain calcium homeostasis in the body [414].

Maintaining calcium homeostasis is important since changes in calcium levels may result in various diseases such as osteoporosis, cardiovascular disease, gastrointestinal diseases, high blood pressure, diabetes mellitus, neurodegenerative diseases, and kidney stones [415,416]. It has been found that NPs may induce an increase in intracellular calcium and disrupt Ca homeostasis [417].

Calmodulin (a calcium-modulated protein) senses the intracellular calcium level, undergoes conformational change upon binding to calcium ions, and can then bind to many calcium sensitive enzymes and target proteins. Increased calmodulin protein expression has been found in human embryonic stem cell-derived glutamatergic neurons treated with Ag NPs, along with increased intracellular calcium levels and oxidative stress [305]. In addition to binding to intracellular calcium, calmodulin has also been found to be able to bind reversibly to calcium fluoride (CaF_2_) NPs [418]. Calmodulin activates myosin light chain kinase (MLCK), which phosphorylates the myosin regulatory light chain allowing myosin ATPase in the myosin head to bind to the actin microfilament and actomyosin contraction to occur [419]. Actin microfilaments are part of the cell’s cytoskeleton and actomyosin contraction is integral to regulating the permeability of the endothelial cell monolayer that functions as a barrier between the blood stream and the surrounding tissue [420]. Treatment of HUVECs with Au NPs, TiO_2_ NPs, SiO_2_ NPs, and polystyrene NPs all resulted in a rise in intracellular calcium, actin rearrangement and alignment, and increased permeability of the endothelial cell monolayer. The increased calcium level may have activated MLCK and led to the observed endothelial barrier dysfunction [420].

A summary of effects of various nanoparticles on zinc-dependent, copper-dependent, iron-dependent, and calcium-dependent proteins is shown in Table 7.

## 8. Effects of Nanoparticles on Detoxification Enzymes

### 8.1. Effects of Nanoparticles on Phase I Detoxification Enzymes

Cytochrome P450 (CYP450) are heme-containing enzymes that play a critical role in the metabolism of xenobiotics through the catalyzation of oxidation reactions [17,422,423]. The main CYP isoforms in humans are CYP1, CYP2, and CYP3, which include CYP1A2, CYP2C9, CYP2D6, and CYP3A4 [423,424]. Many NPs affect the activity of CYPs in vitro and in vivo; however, the mechanisms for how NPs influence CYP function and activity are not well understood [17,422,425].

Several studies have found that NP exposure increases the enzyme activity of various CYPs. A solution of 0.01–1% chitosan NPs induced a dose-dependent increase in CYP3A4 activity in bi-potential human liver (BHAL) cells [426]. Diamond NPs and graphite NPs at concentrations of 3.125–100 mg/L inhibited the activity of CYP1A2, CYP2D6 and CYP3A4 in a microsomal model [427]. Camptothecin encapsulated PLGA NPs at 6.25–100 μg/mL induced a dose-dependent decrease in CYP3A4 activity in HepG2 cells [423]. It has been proposed that Au NPs may interfere with the conformation and activity of CYP2B1 through electrostatic repulsion and may also interfere with the electron transfer in reactions involving CYP [428].

An in silico molecular docking predictive model using Ag_3_ clusters to model Ag NPs found that the modelled Ag_3_ clusters bound to Leu362, Ile362, and Val370 on the active sites of CYP2C9, CYP2C19, and CYP2D6, respectively, and did not interact with CYP1A2, CYP2E1, or CYP3A4 [424].

### 8.2. Effects of Nanoparticles on Phase II Detoxification Enzymes

Glutathione-S-transferase is an enzyme that plays a crucial role in the detoxification of xenobiotics and protection against oxidative stress. Glutathione-S-transferase catalyzes the conjugation reaction between glutathione and the target compounds, allowing it to be removed and excreted [429,430].

Some studies have found that NPs increase GST activity in rat, mice, and human cell lines [431,432]. It was found that ZnO NPs bind to GST but do not impair GST function or conformation [430]. In fact, diabetic rats treated with ZnO NPs for 30 days had increased GST activity in testicular tissue [432]. Interestingly, Ag NPs increased the levels of GST in mice with induced liver cancer [431]. However, amorphous Si NPs decreased GST activity [433].

### 8.3. Effects of Nanoparticles on Metallothioneins

Metallothioneins (MTs) are a family of low molecular weight, metal-binding, and cysteine-rich proteins with various important biological functions. Metallothioneins bind to heavy metals and protect the body from toxicity and oxidative stress and are also involved in maintaining zinc and copper homeostasis [47,379,434,435,436]. The four main isoforms of MTs are MT-1, MT-2, MT-3, and MT-4, where expression levels of each vary depending on the tissue type [351,437].

Various metal NPs may induce the expression of MTs in various cell lines, animal models, and plant models [435,438,439,440,441]. Using UV-vis spectroscopy and TEM, Ag NPs were found to adsorb onto MT-1 [442]. However, in mouse macrophage J774.1 cells, Ag^+^ ions from silver nitrate (AgNO_3_) adsorbed to MT while the Ag NPs themselves did not [443]. Using TEM, DLS, UV-vis, and CD, Ag^+^ ions released from Ag NPs were able to replace the native metals on MT-1 [442].

It is proposed that quantum dots interact with MTs through the binding of thiol groups on the MTs to the metal cores of the quantum dots and/or through electrostatic interactions between the MTs and the quantum dots [46,47]. Coated and uncoated CeO_2_ NPs have been found to interact electrostatically with the MT thiol groups and form a complex which results in complete metal unloading from the MTs, disrupting metal homeostasis [444]. Metallothionein mediated zinc homeostasis was disrupted by copper and Ag ions. Copper ions released from CuO NPs were able to bind to MT-1 in hepatocytes, replacing the zinc that was bound to MT-1 [445]. Silver ions also have a higher binding affinity to MTs than zinc ions and are able to displace the zinc ions that are bound to MTs [446].

A summary of effects of various nanoparticles on detoxification enzymes is shown in Table 8.

## 9. Conclusions

It is important to understand the potential effects of various NPs on biological systems as they are currently being utilized by industry today. Identification of interactions between NPs and specific proteins will provide insight into the pathways and processes that may be affected by their exposure. Of a size that can readily enter cells, either through direct absorption or through phagocytosis, NPs will have either a positive or a negative effect on stress-response systems. In general, NPs have been found to disrupt stress response pathways and induce stress, including oxidative stress response, inflammatory response, mitochondrial function, and stress signaling response pathways. NPs have been found to disrupt the function of antioxidant enzymes. Exposure to various NPs may also cause mitochondrial dysfunction, disrupting cellular homeostasis and leading to induction of apoptosis. Exposure to various NPs has been observed to cause mitochondrial dysfunction, increase expression of mitochondrial fission proteins, decrease expression of mitochondrial fusion proteins, decrease mitochondrial membrane potential, impair oxidative phosphorylation, disrupt cellular homeostasis, and induce mitochondrial mediated apoptosis, with the exceptions being NPs with intrinsic antioxidant properties. However, in some instances, NPs have been proven to be beneficial to the response, including having antioxidant and anti-inflammatory effects. NP treatment may have the effect of “calming the immune system” by inducing the expression of anti-inflammatory cytokines and by inhibiting the expression of pro-inflammatory cytokines. Of the NPs that have these beneficial effects, the Zn- and Ag-containing NPs have been the most studied. NPs have been found to influence several signaling pathways, including insulin, calcium, and ^●^NO signaling. Certain NPs, most of which have antioxidant properties, have been found to be beneficial in activating the insulin signaling pathway, regulating blood glucose levels, and decreasing insulin resistance in diabetic models.

Many NPs have the ability to interact, either directly or indirectly, with various enzymes and proteins, leading to either activation or dysfunction. Nanoparticles have been shown to affect metal homeostasis, including zinc, copper, iron, and calcium homeostasis in biological systems. NPs may also interact with enzymes involved in detoxification including CYPs, GST, and metallothioneins. It is studies that examine these interactions that are currently lacking. Since NPs have been, and are currently, used in industry without strict regulatory guidelines for their use, studies of their effects on stress responses will be critical. NPs are also actively being investigated for their potential in medical applications and in the treatment of various diseases, making it important to further investigate and understand the mechanisms by which NPs interact with proteins and affect signaling pathways involved in stress. NPs are a part of our daily lives now and knowledge of how they impact our stress response is important in their continued use.

## Figures and Tables

**Figure 1 ijms-23-07962-f001:**
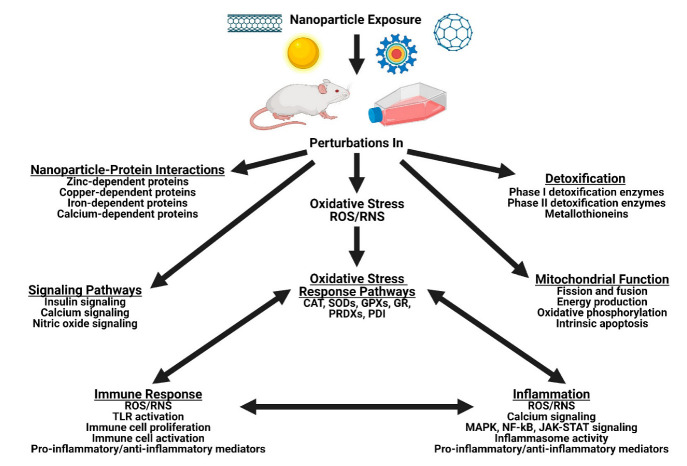
Nanoparticle exposure on biological systems may induce oxidative stress and affect the proteins involved in the stress response pathways (such as CAT, SODs, GPXs, GR, PRDXs, and PDI), the immune response (through TLR activation, immune cell proliferation and activation, and the production of inflammatory mediators), inflammatory responses (through inflammasome activity, the production of inflammatory mediators, and calcium, MAPK, KF-κB, and JAK-STAT signaling), mitochondrial function (including mitochondrial fission and fusion, amount of ATP produced, oxidative phosphorylation, and intrinsic apoptosis), biological signaling pathways (such as insulin, calcium, and nitric oxide signaling), detoxification (involving the Phase I and II detoxification enzymes and metallothioneins), as well as interactions of NPs with zinc-dependent proteins, copper-dependent proteins, iron-dependent proteins, and calcium-dependent proteins.

**Figure 2 ijms-23-07962-f002:**
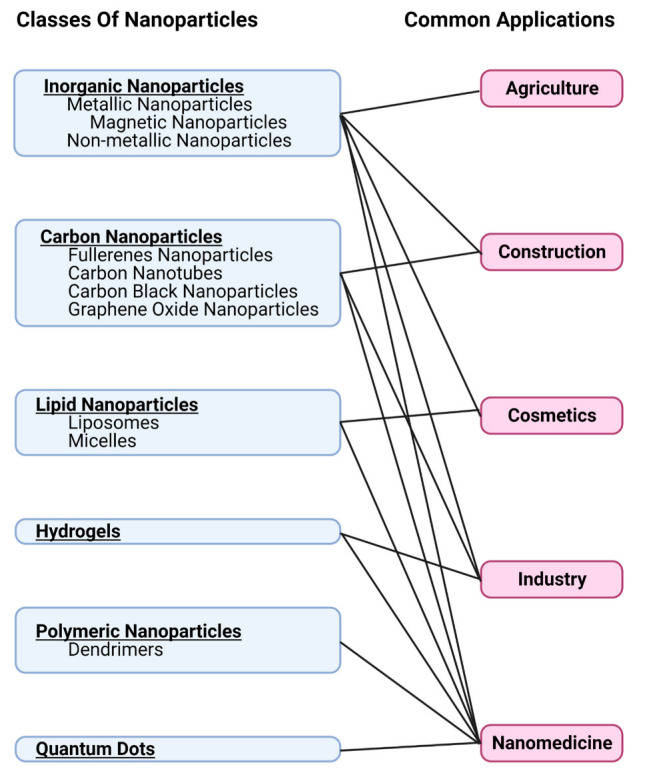
Nanoparticle classes and their applications.

**Table 1 ijms-23-07962-t001:** Applications for nanoparticles used in the studies in this review.

Type of Nanoparticle	Common Applications	References
Ag	Industry, nanomedicine, cosmetics, agriculture	[24,25,26,27,28]
Al and Al_2_O_3_	Industry, nanomedicine, construction	[13,24,33,34]
Au	Industry, nanomedicine	[11,13,24,30,31,32]
Carbon nanotubes	Industry, construction	[13,33,34]
Carbon black	Industry, cosmetics	34,35,36]
CeO_2_	Industry, nanomedicine	[35,36]
Cu and CuO	Industry, nanomedicine, construction	[24,33,34]
Dendrimers	Nanomedicine	[11,13]
Fullerenes	Nanomedicine	[13]
Graphene oxide	Nanomedicine	[37]
Iron oxide	Nanomedicine	[11,13,24]
Liposomes	Nanomedicine, cosmetics	[11,13,34]
Nanoplastics and microplastics	Industry	[38]
Pt	Industry, nanomedicine	[25,26]
Polymers	Nanomedicine	[10,39]
Si and SiO_2_	Industry, nanomedicine, cosmetics, construction, agriculture	[13,17,24,27,28,38,40,41]
TiO_2_	Industry, construction, cosmetics, nanomedicine	[13,17,27,28,39,42,43,44,45]
Quantum dots	Nanomedicine	[11,13,38,46,47]
Zinc oxide	Industry, cosmetics, agriculture	[24,28,48]

**Table 2 ijms-23-07962-t002:** Nanoparticle effects on oxidative stress and stress response pathways.

Type of Nanoparticle	Experimental Model	Protein(s) Affected	Main Findings	References
Ag	n/a	CAT and SOD	Conformational changes to CAT resulting in loss of catalytic activity, but minimal effects to SOD shape and activity	[69]
Ag	HaCaT and A549	Thioredoxin reductase	Decreased expression of selenoproteins	[70]
Ag and Ag^+^	HepG2 and human hepatocytes	PRDXs, GST, myosin, elongation fac-tor 1, 60S ribosomal protein, and 40S ribosomal protein	Direct binding	[71]
CdTe quantum dots	n/a	GPx3	Direct binding through Van der Waals’ forces and hydrogen bonding, resulting in structural changes with increased alpha helical content	[72]
CdTe quantum dots	n/a	GPx3	Interactions with glutamate 136, phenylalanine 132, proline 130, and valine 129 in the GPx3 active site	[73]
4Cu and CuO	RAW264.7	PRDX1, PRDX2, PRDX3, and PRDX6	Increased protein levels of the oxidized form of PRDX1 and the native form of PRDX6, with no change in the levels of PRDX2 and PRDX3	[74]
r/aTiO_2_, rTiO_2_ silica-coated, rTiO_2_ alumina-coated, aTiO_2_, and mwCNT	Human lung epithelial cells and human monocyte-derived macrophages	PRDXs	Association of the nanoparticles with PRDXs	[75]
Selenium-Lovastatin	Female albino rats	Se-dependent GPx	Increased enzyme activity	[76]
TiO_2_	n/a	CAT and SOD	Conformational and functional changes with an increase in alpha helical content and increased exposure of hydrophobic regions	[77]
Pb^2+^, Hg^2+^, Cd^2+^, Fe^3+^, Cu^2+^, Al^3+^	n/a	Human erythrocyte GR	Competitively inhibited by Pb^2+^, Hg^2+^, Cd^2+^, Fe^3+^; and non-competitively inhibited by Cu^2+^ and Al^3+^	[78]
ZnO	Male C57BL/6 mouse liver	PDI-3	Increased PDI-3 gene expression	[79]
ZVFe	Human lymphocytes	Hb	Heme displacement and degradation, and induction of protein carbonylation	[80]

**Table 3 ijms-23-07962-t003:** Effects of nanoparticles on the immune response.

Immune Response	Type of Nanoparticle	Experimental Model	Main Findings	References
Induced Immune Response	Ag	Male NMRI mice	Induced B-cell proliferation	[103]
	Ag NP infused hydrogel	Diabetic rat model	Induced T-cell and macrophage proliferation and significantly decreased wound healing time	[104]
	Ag	RAW 264.7	Increased the expression of TLR-3 with no change in expression of TLR-4	[105]
	Ag_2_O	THP-1	Increased TLR-6 gene expression	[106]
	Al	Male ICR mice	Oxidative damage, neutrophil dysfunction, and increased expression of TNF-α, IFN-γ, IL-1α, IL-1β, IL-2, IL-6, and IL-10	[107]
	ALOOH	THP-1	Increased TLR-4 and TLR-6 gene expression	[106]
	Ce	Cytotoxic T-cells	Activation of cytotoxic T-cells and increased killing activity through decreased ROS, increased NF-κB signaling, increased cytokine production (IL-2 and TNF-α), and increased production of granzyme B and perforin	[108]
	CuO	Female mice	Proliferation and cytokine production by the T-cells and B-cells that changed over time	[109]
	CuO	THP-1	Increased TLR-4 and TLR-6 gene expression	[106]
	Iron oxide	THP-1	Increased TLR-6 gene expression	[106]
	Superparamagnetic iron oxide	Mouse macrophages	Activated TLR-4 receptors, increased the expression of inflammatory cytokines, triggered the translocation of Nrf2 to the nucleus, stimulated autophagy, and increased the gene expression of SR-AI	[110]
	TiO_2_	THP-1	Increased TLR-4 and TLR-6 gene expression	[106]
	ZnO	THP-1	Increased TLR-4 and TLR-6 gene expression	[106]
No Change or Decreased Immune Response	Ag	Rats	Increased T-cell, B-cell, and natural killer cell proliferation; decreased natural killer cell activity; and decreased production of IFN-γ in the spleen	[111]
	Au	Female C57BL/6J mice	No change in lymphocyte cell population, however, increased T-cell population was observed in mice that were pre-sensitized to Au	[112]
	Carbon nanotubes (amino-functionalized)	PBMCs and C57Bl/6 mice	Decreased monocyte and macrophage cell populations in PBMCs, but not in C57Bl/6 mice	[113]
	Graphene	J774	Decreased the gene expression of TLR-5 and decreased the production of IL-1β and IL-6	[114]
	Multi-wall carbon nanotubes	J774	Decreased the gene expression of TLR-5 and decreased the production of IL-1β and IL-6	[114]
Attenuated an Induced Immune Response	Ag	Rats	Decreased the level of IgG induced by KLH	[115]
	Ag and Au	Murine splenic lymphocytes and human blood lymphocytes	Decreased immune cell proliferation stimulated by LPS, Concanavalin A, phytohemagglutinin, or pokeweed mitogen. No change occurred in the unstimulated cells.	[116]
	Superparamagnetic iron oxide	RAW264.7	Attenuated the immune response induced by LPS with decreased expression levels of TLR-4 and NOS, and decreased levels of IL-6 and TNF-α.	[117]

**Table 4 ijms-23-07962-t004:** Summary of the effects of various nanoparticles on inflammation in different experimental models.

Anti-Inflammatory or Pro-Inflammatory	Type of Nanoparticle	Experimental Model	Main Findings	References
Anti-inflammatory	Ag	RAW264.7 cells	Inhibition of pro-inflammatory cytokines	[165,170]
	Ag	BMDMs from C57/BL6 mice	Decrease of NLRP3 inflammasome levels	[172]
	Au	Rats	Inhibition of production of pro-inflammatory cytokines, induction of IL-10	[149]
	Au	Mice	Decreased levels of NF-κB and IL-1β	[199]
	Au	RAW264.7 cells	Suppression of pro-inflammatory cytokines, inhibition of the activation of the JAK-STAT pathway	[150]
	Au	LPS-induced Mouse primary microglia	Decreased production of inflammatory cytokines, mediators, and downregulation of the LPS-induced NF-κB, MAPK, and JAK-STAT signaling pathways	[160]
	Au	LPS-induced RAW264.7 cells	Decreased LPS-induced inflammatory mediators, inhibition of LPS-induced activation of the MAPK and NF-κB signaling pathways	[218]
	Ce	Human dendritic cells	Increased production of anti-inflammatory cytokine IL-10	[136]
	Gadofullerene	Diabetic mice	Decreased expression of pro-inflammatory cytokines (NF-κB, TNF-α, IL-6, and IL-1β)	[154]
	Gadofullerene	Kupffer cells	Decreased LPS-induced inflammatory cytokines	[213]
	Se	Type 2 diabetes rat	Decreased expression of pro-inflammatory cytokines (TNF-α, IL-6, and IL-1β)	[156]
	Se	RAW264.7 cells	Decreased gene expression of pro-inflammatory cytokines (IL-1 and TNF-α), increased gene expression of anti-inflammatory cytokine (IL-10), inhibition of NF-κB and MAPK pathways	[156]
Pro-inflammatory	Ag	HepG2	Activation of NLRP3 inflammasome	[177]
	Ag	Human Monocytes	Activation of NLRP3 inflammasome	[176,179,219]
	Ag	Human keratinocytes	Activation of NLRP3 inflammasome	[166]
	Ag	RAW264.7 cells	Activation of NF-κB pathway	[181]
	Ag	THP-1 cells	Induce formation of pro-inflammatory chemoattractant, 5-HETE	[183]
	Au	Primary neonatal human epidermal keratinocytes	Increased production of pro-inflammatory cytokines, IL-1β, IL-1α, and IL-8	[132]
	Carbon (graphene)	RAW264.7 cells	Increased secretion of TNF-α	[205]
	Carbon (graphene)	BALB/c mice (in vivo and in vitro)	Activation of NF-κB pathway, increased production of pro-inflammatory cytokines	[201]
	Carbon (graphene)	Primary mouse macrophages	Activation of NF-κB pathway	[212]
	SiO_2_	NCI-H292 cells	Increase of production of pro-inflammatory cytokines (TNF-α, IL-6, and IL-8)	[130,133]
	SiO_2_	HUVECs	Activation of MAPK signaling	[200]
	SiO_2_	Huh7 cells	Induction of MAPK genes and increased expression of pro-inflammatory cytokines, TNF-α and IL-8	[158]
	SiO_2_	Ovalbumin-induced asthmatic mice	Activation of NLRP3 inflammasome, increased number of immune cells, inflammatory cytokine production (IL-1β, IL-6, and TNF-α), inflammation of airway	[130]
	SiO_2_	NCI-H292 cells	Activation of NLRP3 inflammasome, increase in gene expression of pro-inflammatory cytokines (IL-1β, IL-6, and TNF-α)	[130]
	SiO_2_	Mouse dendritic cells	Activation of NLRP3 inflammasome	[197]
	TiO_2_	Male Wistar rats	Concentration dependent activation of M1 macrophages	[131]
	TiO_2_	Human macrophages	Induction of IL-6 and high dose resulted in induction of IL-1β and IL-10	[188]
	TiO_2_	K562 cells	Activation of NF-κB pathway	[193]
	TiO_2_	RAW264.7	Activation of NF-κB pathway	[181]
	TiO_2_	Human dermal fibroblasts	Activation of NF-κB pathway	[195]
	TiO_2_	HUVECs	Activation of NF-κB pathway, increased ROS and ^●^NO production	[199]
	TiO_2_	Human dendritic cells	Activation of NLRP3 inflammasome	[136]
	TiO_2_	THP-1 cells	Activation of NLRP3 inflammasome	[186]
	TiO_2_	Murine dendritic cells	Activation of NLRP3 inflammasome	[197]
	TiO_2_	C57BL/6 mice	Induction of pro-inflammatory cytokine production	[202]
	ZnO	THP-1 cells	Activation of NF-κB and MAPK pathways	[137]
	ZnO	Macrophages from female BALB/c mice	Activation of MAPK and NF-κB pathways	[101]
	ZnO	Primary mice dendritic cells	Increased production of pro-inflammatory cytokines IL-6 and TNF-α	[214]

**Table 5 ijms-23-07962-t005:** Effects of nanoparticles on mitochondrial function.

Mitochondrial Function	Type of Nanoparticle	Experimental Model	Main Findings	References
Intrinsic Apoptosis	Ag	HepG2	Activation of intrinsic apoptosis	[230]
	Ag	HCT116	Activation of intrinsic apoptosis	[231]
	Ag	MCF-7	Activation of intrinsic apoptosis	[232]
	Ag	HT22	Intrinsic apoptosis alleviated with selenium	[241]
	Au	HT-29	Activation of intrinsic apoptosis	[237]
	Au nanostars	MCF-7	Activation of intrinsic apoptosis	[238]
	Au nanoprisms	MEFs transformed with SV40	Activation of intrinsic apoptosis	[239]
	Cu	Female Sprague-Dawley rats	Intrinsic apoptosis in the uterine tissue	[242]
	Cu	Swiss albino male mice	Oxidative stress, intrinsic apoptosis, and extrinsic apoptosis in the kidney tissue	[243]
	Cu	A-357	Activation of intrinsic apoptosis	[229]
	Fullerene derivative C_60_(C(COOH)_2_)_2_	HeLa	Alleviated stimulated Cyt c release, intrinsic apoptosis, and mitochondrial membrane potential decrease	[244]
	Nanoplastic polystyrene	A549	Activation of intrinsic apoptosis	[52]
	N-Succinyl-chitosan	K562	Activation of intrinsic apoptosis	[236]
	Phloretin loaded chitosan	Human oral cancer cells	Activation of intrinsic apoptosis	[235]
	SiO_2_	SH-SY5Y	Activation of intrinsic apoptosis	[21]
	Superparamagnetic iron oxide	PC12	Activation of intrinsic apoptosis	[233]
	Quantum dots (cadmium sulphide)	HepG2	Activation of intrinsic apoptosis	[234]
	ZnO	HAECs	Intrinsic apoptosis alleviated with alpha-lipoic acid	[240]
Fission and Fusion	Ag	Sprague-Dawley rats	Increased fission (protein expression of Drp1, p-Drp1, and Fis1 increased) and decreased fusion (protein expression of Opa1 and Mfn2 decreased)	[224]
	Ag	iPSCs	Increased fission and decreased fusion	[245]
	Ag	HepG2	Increased fission, decreased fusion, and decreased mitochondrial biogenesis (protein expression of PGC-1α decreased)	[222]
	Au (combined with TRAIL)	NSCLC cells in female BALB/c mice	Increased fission (increased Drp1)	[246]
	Carbon black FW2	HPAECs	Increased oxidative stress, fission, and intrinsic apoptosis	[49]
	Si	HUVECs	Increased oxidative stress, fission (increased Fis1, Drp1), and intrinsic apoptosis; decreased fusion (decreased Opa1)	[247]
Energy Production	Ag	HepG2, HeLa, A498, PC3, HEK293T	Mitochondrial dysfunction, decreased ATP production, decrease in oxidative phosphorylation, and increase in glycolysis	[253]
	Ag	HSC-3	Mitochondrial depolarization	[268]
	Ag	PANC-1	Decreased mitochondrial membrane potential	[261]
	Ag	Mitochondria from rat liver	Mitochondrial swelling, decreased mitochondrial membrane potential, and uncoupling of oxidative phosphorylation	[262]
	Ag	Male Wistar rats	Decrease in oxidative phosphorylation and increased ROS	[255]
	Al	Mitochondria from male Wistar rat brain tissue	Increased ROS mainly through inhibition of coenzyme Q-cytochrome c oxidoreductase (complex III) and induced intrinsic apoptosis	[259]
	Au	HSC-3	Decreased ATP	[268]
	Carbon nanotubes (single walled and multiwalled)	F10 cells transplanted in a C57 mouse	Decreased mitochondrial membrane potential, decreased succinate dehydrogenase (complex II) activity, and increased intrinsic apoptosis	[256]
	Fe_2_O_3_	Male albino mice	Cardiac dysfunction with damaged mitochondria, decreased ATP production, increased ROS, and intrinsic apoptosis	[263]
	Fe_2_O_3_	F10 cells transplanted in a C57 mouse	Decreased mitochondrial membrane potential, decreased succinate dehydrogenase (complex II) activity, and increased intrinsic apoptosis	[256]
	SiO_2_	U87	Decreased cytochrome c oxidase (complex IV) subunit II protein expression	[258]
	Ti	Male Wistar rats	Decrease in oxidative phosphorylation and increased ROS	[255]
	TiO_2_	Human lymphocytes	Decreased succinate dehydrogenase (complex II) activity	[254]
	TiO_2_	WI-38	Decreased mitochondrial membrane potential	[43]
	ZnO	A549	Decreased cytochrome c oxidase (complex IV) and F_1_F_0_-ATPase (complex V) activity	[257]

**Table 6 ijms-23-07962-t006:** Summary Table of effects of nanoparticles on insulin, calcium, and nitric oxide signaling pathways.

Signaling Pathway	Type of Nanoparticle	Experimental Model	Main Findings	References
Insulin	Ag	Male Sprague Dawley rats	Increased levels of IRS-1, Akt, GSK3β	[288]
	Ag	Male albino rats	Reduction of blood glucose, and increase in serum insulin, glucokinase activity, expression of insulin, insulin receptor, GLUT-2 and glucokinase genes	[277]
	Au	RIN-5F cells	Increased insulin secretion, decreased apoptosis, and increased gene and protein expression of IRS-1, GLUT2, and the insulin receptor	[284]
	Gadofullerene (functionalized with β-alanine)	Diabetic mice	Activated IRS2/PI3K/Akt signalling pathway, normalized blood sugar levels	[154]
	Ni	Rat testicular Sertoli-germ cells	Upregulation of the gene and protein expression of IGFBP-3	[272]
	Platinum	HepG2 cells	Increased activation of Akt	[26]
	Se	Diabetic male Wistar rats	Normalized fasting blood glucose level, reduction of insulin level, normalized levels of phosphorylated: IRS-1, Akt, GSK-3β, and AMPK	[155]
	Superparamagnetic iron oxide	Diabetic male Wistar rats	Normalized fasting blood glucose levels, reduction of insulin level, increased levels of P13K and GSK-3β resulting in increased glucose uptake and decreased insulin resistance	[275]
	TiO_2_	Male CD-1 mice	Increase in blood glucose level, increased IRS-1 phosphorylation, and decreased phosphorylation of Akt	[289]
	ZnO	Male Sprague-Dawley rats	Decreased serum level of IGF-1	[270]
	ZnO	Streptozotocin-induced diabetic male albino rats	Normalized fasting blood glucose, insulin, and insulin resistance score levels, and improved diabetic vascular complications	[286,287]
	ZnO	Male albino rats	Reduction of blood glucose, and increase in serum insulin, glucokinase activity, expression of insulin, insulin receptor, GLUT-2 and glucokinase genes	[39]
	ZnO	Obese C57BL/6 mice	Decreased insulin resistance	[279]
Calcium	Ag	SH-SY5Y	Induced release of calcium from ER	[295]
	Ag	SH-SY5Y, Sprague-Dawley rats	Increased intracellular calcium level, activation of CaMKKβ and AMPK	[298]
	Ag	Zebrafish embryos	Downregulation of calcium signaling	[301]
	Ag	Rat CGCs	Increased intracellular calcium level	[296]
	Ag	Human embryonic stem cell-derived glutamatergic neurons	Decreased expression of NMDA receptor subunits	[305]
	Ag	Male Wistar rats	Increased gene expression of NMDA receptor subunits NR1, NR2A, and NR2B in the brain tissue	[306]
	Carbon	PASMCs	Increased intracellular calcium level	[297]
	Carbon black	Human pulmonary artery endothelial cells	Increased intracellular calcium level	[49]
	CuO	RCC	Increased intracellular calcium level	[299]
	NiO	BEAS-2B cells	Increased intracellular calcium level	[300]
	MgO	Male Wistar rats	Decreased level of glutamate, increased gene expression of NMDA subunit NR2B in the hippocampus	[303]
	Si	HUVECs	Increased intracellular calcium level, decreased activity of Ca^2+^-ATPase, Ca^2+^/Mg^2+^-ATPase, and Na^+^/K^+^-ATPase	[347]
	SiO_2_	PASMCs	Increased intracellular calcium level	[52]
	TiO_2_	PASMCs	Increased intracellular calcium level	[52]
Nitric Oxide	Ag	Rat smooth muscle tracheal rings	Increased iNOS protein expression, ^●^NO production, activation of muscarinic receptors	[310]
	Ag	ASM cells from male Wistar rats	Increased iNOS expression, ^●^NO production, and activation of muscarinic receptors	[311]
	Ag	SVEC4-10 mouse endothelial cells and hFOB 1.19 cells	Increased iNOS expression and ^●^NO production	[313,348]
	Ag	Human embryonic stem cell-derived glutamatergic neurons	Increased iNOS and nNOS expression	[305]
	Ag	Chicken sertoli cells	Increased gene expression of iNOS and eNOS	[314]
	Ag	LPS-induced RAW 264.7 cells	Decreased ^●^NO and iNOS expression	[168]
	Ag	Isolated perfused hearts from rats	Increased ROS, decreased ^●^NO production, decreased iNOS and eNOS expression, but increased expression of phosphorylated eNOS	[315]
	Ag	C57BL/6J male mice fed high-fat Western diet	Decreased iNOS gene expression	[317]
	Au	RAW 264.7 cells	Decreased ^●^NO and iNOS expression	[150,168,218,318]
	Au	Mouse BV-2 microglial cells	Decreased ^●^NO and iNOS expression	[160]
	Au	Rhesus macaque choroid-retinal endothelial RF/6A cells	Decreased VEGF-induced eNOS phosphorylation	[312]
	Carbon	Aluminum-induced Alzheimer’s male albino Wistar rats	Normalized iNOS expression	[334]
	Carbon (graphene QDs)	HepG2 cells	Decreased ^●^NO production	[335]
	Carbon (multi-walled nanotubes)	RAW 264.7 cells	Increased iNOS expression	[342]
	Chitosan	Male albino Wistar rats induced with carcinoma	Normalized ^●^NO level	[344]
	Se (capped with *Ganoderma lucidum* polysaccharides)	LPS-induced RAW 264.7 cells	Normalized ^●^NO level	[156]
	CeO_2_	Male Sprague Dawley rats with LPS-induced sepsis and Kupffer cell macrophages	Decreased iNOS gene and protein expression and ^●^NO production	[36,213]
	CeO_2_	Male Sprague Dawley rats	Decreased ^●^NO production	[35]
	Cu	Wistar albino rats with induced myocardial infarction	Normalized serum ^●^NO level	[331]
	Cu	Male Sprague-Dawley rats	Increased levels of ROS, iNOS, and ^●^NO in the liver tissue	[332]
	Cu	LPS-induced primary macrophages from C57BL/6 mice	Normalized ^●^NO level	[333]
	Cu	LPS-induced primary mouse macrophages	Normalized ^●^NO level	[74]
	Hydroxyapatite	HUVECs	Decreased ^●^NO production and phosphorylated eNOS expression	[345]
	MnO_2_	Cytokine-challenged cartilage explants	Decreased ^●^NO level	[343]
	Propylene glycol alginate sodium sulfite	Streptozotocin-induced diabetic male Wistar rats	Inhibited diabetes-induced decrease in NOS activity and ^●^NO level	[346]
	Pt	LPS-induced RAW 264.7 cells	Decreased ^●^NO and iNOS expression	[327,349]
	Pt	THP-1 cells	Increased ^●^NO level	[25]
	TiO_2_ nanotubes	HUVECs	Downregulation of eNOS transcription factors (KLF2 and KLF4), and decreased ^●^NO level, resulting in decreased eNOS expression	[325]
	Si	Male Wistar rats	Increased ^●^NO level in the liver and kidneys	[338]
	Si	HUVECs	Increased ^●^NO levels, increased iNOS expression, and decreased eNOS expression	[339]
	Si	HUVECs	Decreased ^●^NO level, decreased expression of total NOS and eNOS, and increased iNOS expression	[340]
	Superparamagnetic FeO	LPS-induced RAW 264.7 cells	Increased ^●^NO level	[117]
	Superparamagnetic FeO	RAW 264.7 cells	Decreased iNOS expression	[336,337]
	ZnO	LPS-induced RAW 264.7 cells	Normalized iNOS expression and ^●^NO production	[326]
	ZnO	Streptozotocin-induced diabetic male albino rats	Normalized iNOS expression and ^●^NO production	[287]
	ZnO	Bacterial infection with nontypeable Haemophilus influenzae infected RAW 264.7 cells and C57BL/6 mice	Normalized iNOS expression and ^●^NO production	[350]
	ZnO	Human coronary artery endothelial cells	Increased iNOS and ^●^NO levels	[328]
	ZnO	BALB/c mice macrophages	Increased iNOS and ^●^NO levels	[101]
	ZnO	THP-1 cells	Increased iNOS and ^●^NO levels	[137]
	ZnFe_2_O_4_	Human amnion epithelial cells	Increased iNOS expression	[330]

**Table 7 ijms-23-07962-t007:** Summary of effects of various nanoparticles on zinc-dependent, copper-dependent, iron-dependent, and calcium-dependent proteins in different experimental models.

Protein Interaction	Type of Nanoparticle	Experimental Model	Protein(s) Affected	Main Findings	References
Zinc-dependent	CuO	A549 cells	HDAC	Inhibition of protein activity	[361]
	ZnO	SH-SY5Y cells	ZnT1 and ZIPs	Increased protein expression	[363]
Copper-dependent	Ag	n/a	AChE	Disruption of secondary structure, decrease of protein activity	[421]
	CuO	Renal cell carcinoma cells	Copper chaperones	Regulates the chaperones, resulting in copper transportation disruption	[299]
	SiO2	Drosophila melanogaster	Copper transporters	NPs competitively bound to proteins, resulting in copper deficiency	[372]
Iron-dependent	Ag	n/a	Hb (human)	Induction of slight changes to secondary structure	[399]
	Ag	n/a	Hb (human)	Induction of structural changes	[400]
	Ag	n/a	Hb (bovine)	Induction of changes to both secondary and tertiary structures	[401]
	Al_2_O_3_	n/a	Hb (human)	Induction of changes to quaternary structure	[409]
	Au	n/a	Transferrin	Changes to secondary structure	[373,382]
	CeO_2_	n/a	Hb (human)	Degradation of heme, structural changes	[404]
	CeO_2_	n/a	Hb (human)	Induction of changes to secondary structure	[405]
	Fe (zero valent)	n/a	Hb (human)	Induction of changes to tertiary structure	[80]
	Graphene oxide	n/a	Hb (human)	Degradation of quaternary and secondary structure	[407]
	Iron oxide	n/a	Cyt c	Reduction of heme group in non-PEG coated NPs	[389]
	Magnetic NPs (Fe III and Fe II)	n/a	Lactoferrin	NPs absorbed onto protein	[385]
	NiO	n/a	Hb	Induction of changes to secondary and quaternary structure	[410]
	Manganese	n/a	Cyt c	Induction of conformational changes to tertiary structure	[391]
	Si	n/a	Cyt c	Adsorption of protein to NPs resulting in structural stabilization, however the active site became inaccessible to ligands	[387]
	Si	n/a		NPs absorbed onto Cyt c resulting in conformational changes and modifications to the active site	[388]
	Si	n/a	Hb (porcine and human)	Loss of secondary structure	[20]
	Si	n/a	Hb (bovine)	Structural changes and heme degradation	[395]
	SiO_2_	n/a	Hb (human)	Structural changes, heme degradation, release of iron from Hb	[124]
	SiO_2_	n/a	Hb (human)	Adsorption of protein to NPs	[360]
	ZnO	n/a	Hb (human)	Induction of structural changes	[398]
	TiO_2_	Human erythrocytes	Hb (human)	Induction of structural changes and degradation of heme group	[396]
Calcium-dependent	CaF_2_	n/a	Calmodulin	Irreversible binding	[418]

**Table 8 ijms-23-07962-t008:** Summary of effects of various nanoparticles on detoxification proteins in different experimental models.

Protein Interaction	Type of Nanoparticle	Experimental Model	Protein(s) Affected	Main Findings	References
Phase I Detoxification Enzymes	Diamond and graphite	Microsomal based models	CYP1A2, CYP2D6, and CYP3A4	Inhibition of activity	[427]
	Camptothecin encapsulated PLGA	HepG2 cells	CYP3A4	Dose-dependent decrease in activity	[423]
Phase II Detoxification Enzymes	ZnO	Testicular tissue in diabetic rats	GST	Increase in activity	[432]
	Ag	Mice with induced liver cancer	GST	Increase in levels	[431]
	Amorphous Si	HepG2 cells	GST	Decrease in activity	[433]
Metallothioneins	Ag	n/a	MT-1	Adsorbed to protein	[442]
	Ag	J774.1 cells, UV-vis and CD	MT-1	Ag^+^ ions released from NPs replaced native metals on MT-1	[443]
	Ag	n/a	MT-1	Ag^+^ ions released from NPs replaced native metals on MT-1	[442]
	CeO_2_	n/a	MT	Interaction with thiol groups	[444]
	CuO	HepG2	MT-1	Cu ions released from NPs replaced Zn bound to MT-1	[445]

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
