# Peer review of "Nanoparticle Effects on Stress Response Pathways and Nanoparticle–Protein Interactions"

_ijms, 2022, doi:10.3390/ijms23147962_

Round 1

Reviewer 1 Report

The topic of this review paper on nanoparticles is a very interesting issue and it has a good structure.

However, I consider that it need a in deep revision before been in condition to be considered for publication.

The main concerns are as follows:

 (1).- ABSTRACT.

The abstract is very unspecific. It seems more like a brief Introduction that an abstract of the manuscript. It is describing also something like a list of issues: “nanoparticles have the potential to either positively or negatively affect… bla, bla, bla…..” Something more conclusive with critical analysis may be expected as a conclusion of the question raised up at the beginning: “it is not clear how nanoparticles may affect stress response pathways and interact with proteins 11 in biological systems

GENERAL APPROACH OF THE MANUSCRIPT

The manuscript contains a lot of descriptions of text, text, and text, just describing issues that are also available in the reported bibliographic references.

The justification of publishing a Review is to offer to the readers something more useful and elaborated, but obviously, it requires a hard-strong effort by the authors. Perhaps might be more useful to focus in fewer types of effects or fewer types of nanoparticles but to analyze them more in deep with more tabulated and graphic information

For that is needed:

(2).- For each section or different topic, I strongly recommend presenting a tabulated summary of the information with a structure order and organization of the selected most relevant studies. For example, a Table with 1 line for each relevant main study (or group of studies) containing the following columns: (/reference/type of nanoparticle/experimental model/main findings-effects/reference(s)/). They may be ordered either by the type of particle or the experimental model or the kind of effects, of a combination.

(3).-I also recommend including some schemes or figures illustrating the relationship among types of particles and effects.

(4).-Authors has also to do an acritical analysis of the published information and try to do suggestions about the mechanistic pathways from molecular to cellular to clinical observation, where possible, or to conclude with no conclusions. This kind of analysis might also try to be shown under schemes or figures. In this sense the general description illustrated in Figure 2 is fine. However, more specific pathways may be developed for the specific sections.

Other minor issues:

(5).-INTRODUCTION (Section 1).

 Only in the INTRODUCTION, some graphic information is illustrated. This is fine and the figures are OK.

The purpose of the review is not a review about which are the concept and different available nanoparticles. Therefore, a detailed description of the type of nanoparticles is not needed and might be summarized, maintaining the Figure 1, or including the more additional information if the Table format.

(6).-Nanoparticles and neuroinflammation and glia cells toxic effects.

One aspect that I found to be of high interest, and interest for a regulatory purpose, is the section about neurotoxicity, and its relationship with glia and BBB. Probably to extend a little more this section considering the relationship between neurotoxicity and interaction with glial cells. Perhaps authors consider the observation reported by Altunbek and coworkers: (Altunbek et al. Int. J. Biol. Macromol. 2018, 118, 271–278. http://doi.org/10.1016/j.ijbiomac.2018.06.059), and those of Sogorb and collaborators (Fuster, et al (2021) Cells. Int. J. Mol. Sci. 2021, 22, 2084. https://doi.org/10.3390/ijms22042084; Fuster et al (2022) Int. J. Mol. Sci. 2022; 23(4):2267. https://doi.org/10.3390/ijms23042267 )). Fuster et al Toxicol. Appl. Pharmacol. 2020, 404, 115178. http://doi.org/10.1016/j.taap.2020.115178

 (7). Calcium signaling and Ca-dependent protein. I wonder if in relation to the Ca-signaling effect, the implication on renal cells and renal function might be addressed.

Author Response

We thank the reviewers for their constructive comments. We appreciate all of the valuable comments they have provided, which were helpful for revising and improving our manuscript “Nanoparticle Effects on Stress Response Pathways and Nanoparticle-Protein Interactions”. We have carefully considered the reviewer’s comments and have modified the manuscript accordingly. If there are other aspects in the manuscript that require further clarification, kindly let us know and we will revise accordingly.

Reviewer 1

  • The abstract is very unspecific. It seems more like a brief Introduction that an abstract of the manuscript. It is describing also something like a list of issues: “nanoparticles have the potential to either positively or negatively affect… bla, bla, bla…..” Something more conclusive with critical analysis may be expected as a conclusion of the question raised up at the beginning: “it is not clear how nanoparticles may affect stress response pathways and interact with proteins in biological systems”

We have completely rewritten the abstract to be more specific and less “list-like”. We have added a more conclusive and critical analysis of the current knowledge of nanoparticle-protein interactions and their biological significance.

  • The justification of publishing a Review is to offer to the readers something more useful and elaborated, but obviously, it requires a hard-strong effort by the authors. Perhaps might be more useful to focus in fewer types of effects or fewer types of nanoparticles but to analyze them more in deep with more tabulated and graphic information

We have addressed this comment by providing summary tables at the end of each section which provide an overview of the main findings of the section. In many cases, there is conflicting evidence (eg. anti-oxidant versus pro-oxidant, anti-inflammatory versus pro-inflammatory, etc.) for each nanoparticle. We chose not to focus on a limited number of nanoparticles, but instead, focus on general effects of classes of nanoparticles. We hope that the summary tables help readers to form conclusions on the effects of individual classes on nanoparticles on specific stress pathways.

  • For each section or different topic, I strongly recommend presenting a tabulated summary of the information with a structure order and organization of the selected most relevant studies. For example, a Table with 1 line for each relevant main study (or group of studies) containing the following columns: (/reference/type of nanoparticle/experimental model/main findings-effects/reference(s)/). They may be ordered either by the type of particle or the experimental model or the kind of effects, of a combination.

We have added tables, with the headings mentioned by the reviewer, which summarize findings from the most relevant studies. We have ordered these tables by the type of nanoparticle and have used the headings suggested by the reviewer.

  • I also recommend including some schemes or figures illustrating the relationship among types of particles and effects.

We have collectively placed classes of nanoparticles and grouped their effects on pathways in tabular form in lieu of figures. We hope that this illustrates the relationship between the type of particle and the specific effect. If schematics or figures are required, we will create these.

  • Authors has also to do an acritical analysis of the published information and try to do suggestions about the mechanistic pathways from molecular to cellular to clinical observation, where possible, or to conclude with no conclusions. This kind of analysis might also try to be shown under schemes or figures. In this sense the general description illustrated in Figure 2 is fine. However, more specific pathways may be developed for the specific sections.

We chose to do the analysis that the reviewer required in tabular format for each section of the review. In many cases, an illustration would be difficult due to conflicting effects for each nanoparticle present in the literature. We felt that it was best illustrated in the form of tables provided in each section of the review.

  • The purpose of the review is not a review about which are the concept and different available nanoparticles. Therefore, a detailed description of the type of nanoparticles is not needed and might be summarized, maintaining the Figure 1, or including the more additional information if the Table format.

We have shortened the section in the Introduction which provides a detailed description of each nanoparticle and moved that information to Table 1. We hope that this makes the Introduction less wordy, while providing a complete summary of the types of nanoparticles and their descriptions.

  • One aspect that I found to be of high interest, and interest for a regulatory purpose, is the section about neurotoxicity, and its relationship with glia and BBB. Probably to extend a little more this section considering the relationship between neurotoxicity and interaction with glial cells. Perhaps authors consider the observation reported by Altunbek and coworkers: (Altunbek et al. Int. J. Biol. Macromol. 2018, 118, 271–278. http://doi.org/10.1016/j.ijbiomac.2018.06.059), and those of Sogorb and collaborators (Fuster, et al (2021) Cells. Int. J. Mol. Sci. 2021, 22, 2084. https://doi.org/10.3390/ijms22042084; Fuster et al (2022) Int. J. Mol. Sci. 2022; 23(4):2267. https://doi.org/10.3390/ijms23042267 )). Fuster et al Toxicol. Appl. Pharmacol. 2020, 404, 115178. http://doi.org/10.1016/j.taap.2020.115178

We have included new sections on neurotoxicity, involving microglial cells, in Sections 4.1, 4.2, 4.3 and 6.3 of the review. We have included the references suggested by the reviewer and incorporated the information found within into the review.

  • Calcium signaling and Ca-dependent protein. I wonder if in relation to the Ca-signaling effect, the implication on renal cells and renal function might be addressed.

We have added a new section (Section 6.2) on the effects of nanoparticles on calcium signalling. We have also added effects on renal cell function to Section 5.2.

Reviewer 2 Report

In general, the article is very interesting. The work contains a lot of interesting and modern information. The mechanisms of action of nanoparticles are considered comprehensively. However, more illustrations in the form of figures and tables are required, which will improve the perception of information.

Figure 2 legend should be more informative

Effects of Nanoparticles on Oxidative Stress and Stress Response Pathways. This part should be illustrated with a figure or table.

Nanoparticles and the Immune Response. The head is very large. Should be illustrated with a table.

Selenium nanoparticles are completely overlooked in the work. They have pleiotropic properties and, on the one hand, they have an anti-cancer effect, and on the other hand, they protect normal cells from damage during stress. Their effects are carried out, among other things, through the regulation of Ca2+ cell dynamics. The latest data on selenium nanoparticles should be discussed in a review. Ref: https://pubmed.ncbi.nlm.nih.gov/34884629/, https://pubmed.ncbi.nlm.nih.gov/34360564/, https://pubmed.ncbi.nlm.nih.gov/34439975/, https://pubmed.ncbi.nlm.nih.gov/35216476/ , https://pubmed.ncbi.nlm.nih.gov/35110605/

Ref: Solovyev, N.D. Importance of selenium and selenoprotein for brain function: from antioxidant protection to neuronal signalling. J. Inorg. Biochem2015153, 1–12.

Sadek, K.M.; Lebda, M.A.; Abouzed, T.K.; Nasr, S.M.; Shoukry, M. Neuro- and nephrotoxicity of subchronic cadmium chloride exposure and the potential chemoprotective effects of selenium nanoparticles. Metab. Brain. Dis201732, 1659–1673.

Conclusions should be written more concisely. The information in the conclusions can be used as a separate discussion chapter.

Author Response

We thank the reviewers for their constructive comments. We appreciate all of the valuable comments they have provided, which were helpful for revising and improving our manuscript “Nanoparticle Effects on Stress Response Pathways and Nanoparticle-Protein Interactions”. We have carefully considered the reviewer’s comments and have modified the manuscript accordingly. If there are other aspects in the manuscript that require further clarification, kindly let us know and we will revise accordingly.

Reviewer 2

  • In general, the article is very interesting. The work contains a lot of interesting and modern information. The mechanisms of action of nanoparticles are considered comprehensively. However, more illustrations in the form of figures and tables are required, which will improve the perception of information.

The reviewer is correct in that more tabular or illustrative summaries of each section were required in the manuscript. We have added summary tables at the end of each section of the review and we hope that this allows the reader to form some conclusions as to the primary effects of each class of nanoparticles. In many instances, there was conflicting evidence on effects, which would be difficult to illustrate in a figure, so we opted to provide tables instead.

  • Figure 2 legend should be more informative

The reviewer is correct in that Figure 2 legend was too brief and uninformative. We have expanded the legend of Figure 2 to include more information about what is shown in the figure.

  • Effects of Nanoparticles on Oxidative Stress and Stress Response Pathways. This part should be illustrated with a figure or table.

We have provided a summary of this section with a table at the end to outline the effects of each class of nanoparticle on oxidative stress parameters.

  • Nanoparticles and the Immune Response. The head is very large. Should be illustrated with a table.

We have provided a summary of this section with a table at the end to outline the effects of each class of nanoparticle on parameters of the immune response.

  • Selenium nanoparticles are completely overlooked in the work. They have pleiotropic properties and, on the one hand, they have an anti-cancer effect, and on the other hand, they protect normal cells from damage during stress. Their effects are carried out, among other things, through the regulation of Ca2+ cell dynamics. The latest data on selenium nanoparticles should be discussed in a review. Ref: https://pubmed.ncbi.nlm.nih.gov/34884629/, https://pubmed.ncbi.nlm.nih.gov/34360564/, https://pubmed.ncbi.nlm.nih.gov/34439975/, https://pubmed.ncbi.nlm.nih.gov/35216476/ , https://pubmed.ncbi.nlm.nih.gov/35110605/

Ref: Solovyev, N.D. Importance of selenium and selenoprotein for brain function: from antioxidant protection to neuronal signalling. J. Inorg. Biochem. 2015, 153, 1–12.

Sadek, K.M.; Lebda, M.A.; Abouzed, T.K.; Nasr, S.M.; Shoukry, M. Neuro- and nephrotoxicity of subchronic cadmium chloride exposure and the potential chemoprotective effects of selenium nanoparticles. Metab. Brain. Dis. 2017, 32, 1659–1673.

We have included new sections on the effects of selenium nanoparticles in Sections 4.8 and 6.1. The effects cannot be overemphasized enough of this important nanoparticle. We have incorporated the references that the reviewer has suggested into the manuscript, within these new sections.

  • Conclusions should be written more concisely. The information in the conclusions can be used as a separate discussion chapter.

The reviewer is correct in that the Conclusion section of the manuscript was too lengthy and simply summarized the main findings of the review. We have completely rewritten the Conclusions section to shorten it and make it more conclusive. We have removed redundant reiteration of what was already stated in the review to replace it with a more conclusive evaluation of the effects of each class of nanoparticles (also summarized in the tables).

Round 2

Reviewer 2 Report

The authors finalized the article and took into account all my comments. The article may be accepted for publication.